# Self-Augmenting Retrieval for Diffusion Language Models

**Paul Jünger** [1]   **Justin Lovelace** [1]   **Linxi Zhao** [1]   **Dongyoung Go** [1]   **Kilian Q. Weinberger** [1]

## Abstract

Discrete diffusion language models generate text by iteratively denoising an entire response in parallel. At each step, they predict tentative tokens for every masked position, committing the confident predictions to the output and discarding the unconfident ones. We show that the discarded tokens are in fact a useful lookahead signal for retrieval-augmented generation: even low-confidence tokens often surface salient entities early in the denoising trajectory, enabling retrieval of stronger evidence before the output is finalized. We exploit this through Self-Augmenting Retrieval for Diffusion Language Models (SARDI), a dynamic RAG framework that uses these lookahead tokens to guide retrieval during denoising. SARDI is training-free, retriever-agnostic, and applicable to any reasoning-capable discrete diffusion language model. Across five multi-hop QA benchmarks, SARDI outperforms current training-free diffusion and autoregressive retrieval baselines at up to $8\times$ higher throughput. Our code is available at https://github.com/pauljngr/SARDI.

## 1. Introduction

Large language models are increasingly deployed as interfaces to external knowledge, with retrieval-augmented generation grounding responses in retrieved evidence (Lewis et al., 2020; Izacard & Grave, 2021). Yet the dominant decoding paradigm, token-by-token autoregressive (AR) generation, ties latency directly to output length (Pope et al., 2023) and forces retrieval to condition on a left-to-right committed prefix (Trivedi et al., 2023). *Diffusion language models* (DLMs) offer an alternative. Rather than emitting one token at a time, they iteratively denoise a corrupted sequence, updating many positions in parallel at each step

[1]Department of Computer Science, Cornell University. Correspondence to: Paul Jünger <pj287@cornell.edu>.

*Proceedings of the $43^{rd}$ International Conference on Machine Learning*, Seoul, South Korea. PMLR 306, 2026. Copyright 2026 by the author(s).

with a test-time-adjustable number of iterations (Austin et al., 2021; Li et al., 2022; Snell et al., 2025; Ye et al., 2025; Lovelace et al., 2023). Recent discrete DLMs such as DREAM-7B (Ye et al., 2025) and LLaDA (Nie et al., 2026) have matured into competitive models at scale, raising the question of how their non-autoregressive structure can be exploited beyond raw throughput.

Denoising unlocks two structural advantages over AR generation in retrieval-augmented settings: one for retrieval and one for decoding.

**Diffusion Trajectories as Lookahead for Retrieval.** Diffusion language models refine the entire response at once: at every denoising step they produce tentative predictions for *all* token positions. This trajectory of intermediate predictions surfaces salient entities and relations before the output is finalized. This is especially useful for multi-hop question answering, where the evidence needed for later reasoning steps depends on intermediate *bridge* entities that the question alone does not specify. Figure 1 illustrates this: asked which city is home to the museum that displays the Mona Lisa, a static retriever cannot retrieve the passage *"The Louvre is located in Paris"* without first identifying the bridge entity *Louvre*. Intermediate diffusion states surface this entity early in the trajectory, enabling retrieval of the second-hop evidence before the final answer is committed. Diffusion trajectories thus let the model "peek into the future" of its own generation to improve retrieval.

**RAG grounding promotes parallel decoding.** Realizing the speedup potential of parallel decoding has proven challenging in practice: sampling multiple tokens simultaneously risks generating conflicting or incoherent spans, errors that can cascade through the response and degrade output quality (Wu et al., 2026). RAG changes this. When generation is grounded in informative evidence $D$, many output tokens are copied or paraphrased straight from the context and are therefore conditionally independent given $D$. We confirm this empirically in Section 5.5: grounding sharply reduces inter-token dependence, promoting parallel decoding.

We propose **Self-Augmenting Retrieval for Diffusion Language Models (SARDI)**, the first framework to condition retrieval on intermediate diffusion states. SARDI interleaves

**Which city is home to the museum that displays the Mona Lisa?** (2-hop question)

**Step 0** [MASK] [MASK] [MASK] [MASK] [MASK] [MASK] [MASK] [MASK]

**retrieve** *+ doc. retrieval (based on question):* *"The Mona Lisa is displayed at the Louvre [...]"*

**Step 1** Bridge entity

**denoise** [MASK] is home to the Louvre, which displays the Mona Lisa.

**retrieve** *+ doc. retrieval (based on question+partial generation):* *"The Louvre is located in Paris."*

**Output** Paris is home to the Louvre, which displays the Mona Lisa.

*Figure 1.* Static question-only retrieval can fail on multi-hop QA when the question does not specify the bridge entity. Intermediate diffusion states often surface such entities early, enabling retrieval of later-hop evidence before the output is finalized.

retrieval with denoising: at each iteration, it constructs a query from the partially denoised sequence, retrieves fresh evidence, and conditions the next step on the updated context. Central to SARDI is a **separation between retrieval and generation confidence** unique to non-autoregressive decoders: speculative future tokens can inform retrieval long before they are stable enough to commit to the output. SARDI is training-free and works plug-and-play with any discrete diffusion language model that can produce reasoning traces.

We empirically validate two properties that make diffusion language models well suited to retrieval. First, we demonstrate that intermediate denoising states surface bridge entities earlier than autoregressive generation, providing a stronger signal for retrieval. Second, grounding generation in retrieved evidence sharply reduces inter-token mutual information, making the response easier to decode in parallel. Together, this lets SARDI dominate the quality–latency frontier across five multi-hop QA benchmarks, outperforming current training-free diffusion and autoregressive baselines at substantially lower latency.

## 2. Related Work

**Retrieval-augmented generation.** Early RAG systems (Lewis et al., 2020; Izacard & Grave, 2021; Karpukhin et al., 2020; Guu et al., 2020) perform *single-shot* retrieval from the input query and keep the retrieved context fixed throughout generation. This constrains their effectiveness on multi-hop tasks, where the evidence needed for later reasoning steps depends on bridge entities that the question does not name (Yang et al., 2018; Ho et al., 2020). All existing RAG approaches for diffusion language models are restricted to this single-shot paradigm (Yu et al., 2026; Fang et al., 2026).

**Dynamic and agentic retrieval in autoregressive LMs.** For autoregressive models, this single-shot limitation has been addressed by a line of work that interleaves retrieval with generation, deriving follow-up queries from the tokens produced so far (Trivedi et al., 2023; Jiang et al., 2023; Jeong et al., 2024; Asai et al., 2024). Most relevant to this work is FLARE (Jiang et al., 2023), which makes retrieval *forward-looking*: rather than querying only on already-committed text, FLARE first generates a tentative next sentence and, if that sentence contains low-confidence tokens, uses it as a query to retrieve fresh evidence. Then, it regenerates the sentence under the updated context. While anticipating future tokens does help, autoregressive decoding makes this lookahead fragile: the tentative span is generated left-to-right, so a single early error can compound and produce hallucinated queries that retrieve irrelevant documents (see results in Section 5.3). In contrast, SARDI predicts all tentative tokens in parallel, so errors do not compound.

More recently, agentic systems generate explicit search queries via planning and self-reflection (Yao et al., 2022; Asai et al., 2024; Xu et al., 2026; Li et al., 2025; Jin et al., 2025). While effective, these approaches typically require specialized training via reinforcement learning, increasing engineering complexity and computational overhead. In contrast, SARDI is *plug-and-play* and requires no learned retrieval controller or query generator. To the best of our knowledge, SARDI is the first retrieval framework that conditions retrieval on intermediate diffusion states and refreshes evidence throughout the denoising trajectory.

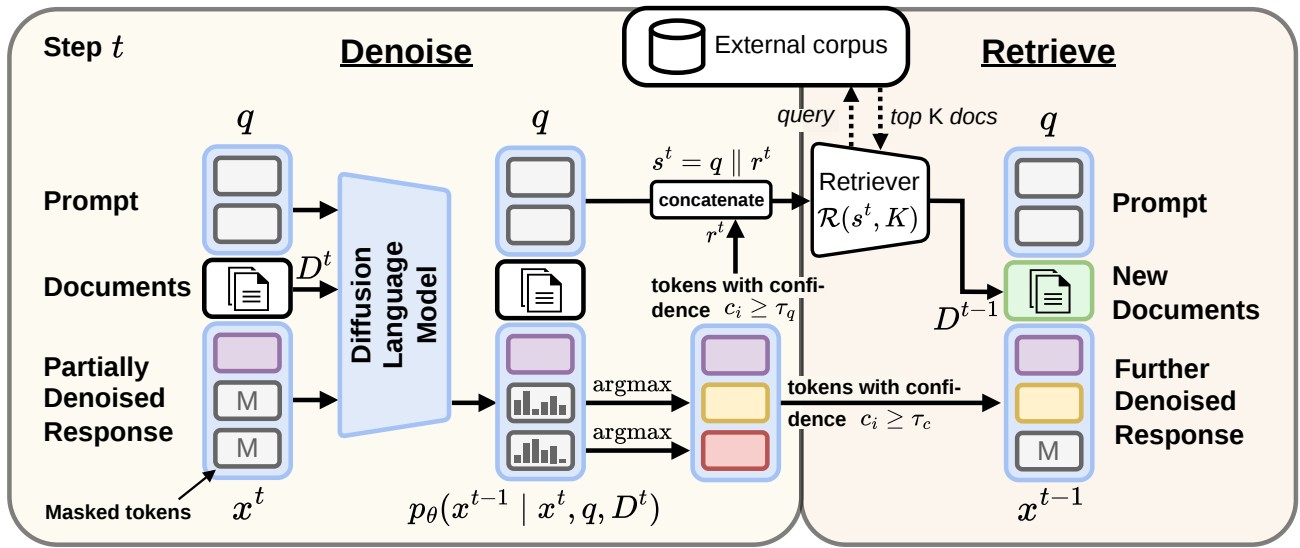

**Figure 2.** Overview of self-augmenting retrieval for diffusion language models. At step $t$, the diffusion LM denoises the partially masked response. Tokens with confidence $c_i \geq \tau_q$ then form a query to refresh the retrieved evidence, while only the more confident tokens ($c_i \geq \tau_c$) are committed to the next response $x^{t-1}$. Speculative tokens can thus inform retrieval before they are stable enough to commit.

## 3. Background

We consider open-domain (multi-hop) question answering. Given a question $q$, the goal is to generate a response $x = (x_1, \ldots, x_L) \in \mathcal{V}^L$ of length $L$ over a finite vocabulary $\mathcal{V}$. Retrieval-augmented generation (RAG) retrieves $K$ passages $D = \{d_1, \ldots, d_K\}$ from an external corpus $\mathcal{C}$ via a query $s$, and conditions generation on both $q$ and $D$.

### 3.1. Autoregressive Language Models

Autoregressive (AR) language models factor the response likelihood left-to-right:

$$p_\theta(x \mid q, D) = \prod_{i=1}^{L} p_\theta(x_i \mid x_{<i}, q, D), \qquad (1)$$

where $x_{<i} = (x_1, \ldots, x_{i-1})$ and $\theta$ denotes the model parameters. Decoding proceeds one token at a time, so retrieval can only condition on the committed prefix $x_{<i}$.

### 3.2. Discrete Diffusion Language Models

Discrete diffusion language models (DLMs) process all positions in parallel through iterative denoising. A DLM defines a sequence of states $\{x^t\}_{t=0}^{T}$, where $x^T = ([\text{MASK}], \ldots, [\text{MASK}])$ is fully masked and $x^0 \in \mathcal{V}^L$ is the final output. Generation proceeds from $t = T$ to $t = 0$: at each step, given a partially masked sequence $x^t \in (\mathcal{V} \cup \{[\text{MASK}]\})^L$, $q$, and $D$, the denoiser predicts a per-position distribution

$$p_\theta(x_i^{t-1} \mid x^t, q, D), \qquad i \in \{1, \ldots, L\}, \qquad (2)$$

and selected masked positions are *unmasked* to their argmax.

**Parallel decoding and RAG.** Sampling positions independently approximates the joint distribution $p_\theta(x \mid q, D)$ by the product of its marginals. This breaks down under strong inter-token dependence. For example, the first name "Albert" makes "Einstein" very likely, whereas independent sampling can produce "Albert Curie". We argue that RAG sharply reduces such dependence: tokens copied or paraphrased from retrieved passages are largely determined by $D$, leaving adjacent positions nearly independent. We validate this hypothesis in Section 5.5, confirming that RAG is a regime well-suited to parallel decoding.

## 4. Self-Augmenting Retrieval for Diffusion

The diffusion trajectory $\{x^t\}_{t=0}^{T}$ exposes a sequence of intermediate states at which retrieval can be revisited. Whereas static RAG issues a single query from the question, *Self-Augmenting Retrieval for Diffusion* (SARDI) refines the query as the response takes shape, surfacing new evidence at every step. SARDI interleaves retrieval with denoising (Figure 2). At each step $t$, the model predicts a token for every masked position with a confidence score $c_i = \max_{v \in \mathcal{V}} p_\theta(v \mid x^t, q, D^t)$. Depending on $c_i$, a position is used to:

(i) **Retrieve.** If $c_i \geq \tau_q$ (the *query threshold*), the predicted token is added to the retrieval query, refreshing the evidence for the next step.

(ii) **Commit.** If $c_i \geq \tau_c$ (the *commit threshold*), the token prediction is committed to; the rest of the tokens are remasked.

Separating the two thresholds is the central design choice in SARDI: because $\tau_q \leq \tau_c$, tentative tokens can inform retrieval well before they are reliable enough to commit. The following sections detail query construction (Section 4.1), evidence refresh (Section 4.2), and confidence-based commitment (Section 4.3).

### 4.1. Query Construction

As denoising progresses, the emerging response surfaces intermediate entities — names, dates, relations — that the question does not contain but that later-hop retrieval needs. We want to feed these to the retriever as early as possible, even while they are still uncertain. This is safe because retrieval and generation tolerate errors very differently: committing an incorrect token can directly corrupt the output, whereas retrieval is robust to noisy queries. We can therefore set the query threshold $\tau_q$ well below the commit threshold $\tau_c$, exposing tentative tokens to the retriever long before they are reliable enough to commit. At $\tau_q = 0$ every position enters the query (maximal lookahead); raising $\tau_q$ restricts it to more confident tokens.

Concretely, let $x^t \in (\mathcal{V} \cup \{[\texttt{MASK}]\})^L$ be the current sequence and define $c_i = \max_{v \in \mathcal{V}} p_\theta(v \mid x^t, q, D^t)$ as the model confidence at position $i$. We form the proxy sequence $\tilde{x}^t$ by using token predictions with confidence at least $\tau_q$:

$$\tilde{x}_i^t = \begin{cases} x_i^t, & x_i^t \neq [\texttt{MASK}] \\ \arg\max_v p_\theta(v \mid x^t, q, D^t), & c_i \geq \tau_q \\ [\texttt{MASK}], & \text{otherwise.} \end{cases} \quad (3)$$

The proxy is detokenized to $r^t = \text{Detokenize}(\tilde{x}^t)$ (remaining $[\texttt{MASK}]$ tokens are dropped), and the retrieval query concatenates the question with this intermediate response:

$$s^t = q \,\|\, r^t. \quad (4)$$

The fixed question $q$ anchors the query when early predictions are noisy, while the evolving $r^t$ progressively specializes retrieval. Unless otherwise specified we use $\tau_q = 0$; Section 5.4 reports an empirical sweep validating this choice.

### 4.2. Evidence Refresh

Using the query $s^t$ constructed above, SARDI retrieves a fresh set of $K$ passages at each step:

$$D^{t-1} \leftarrow \mathcal{R}(s^t, K), \quad (5)$$

where $\mathcal{R}$ is the retriever. The new evidence $D^{t-1}$ replaces the previous context entirely and conditions the next denoising step. We use BM25 (Robertson et al., 2009) in our experiments for efficiency, but SARDI is retriever-agnostic and works with any sparse or dense retriever.

---

**Algorithm 1** SARDI: Self-Augmenting Retrieval for Diffusion Language Models

---

**Require:** Question $q$, retriever $\mathcal{R}$, denoiser $p_\theta$, query threshold $\tau_q$, commit threshold $\tau_c$, context size $K$
**Ensure:** Generated sequence $x$
1: $x \leftarrow ([\texttt{MASK}], \dots, [\texttt{MASK}])$ {Fully masked initialization}
2: $D \leftarrow \mathcal{R}(q, K)$ {Initial retrieval from question only}
3: **while** not fully unmasked **do**

4:     *// — Decode: commit high-confidence tokens —*
5:     Compute $p_\theta(\cdot \mid x, q, D)$ for all masked positions
6:     $c_i \leftarrow \max_{v \in \mathcal{V}} p_\theta(v \mid x, q, D)$ for each masked position $i$
7:     $\mathcal{U} \leftarrow \{i : x_i = [\texttt{MASK}] \wedge c_i \geq \tau_c\}$
8:     **if** $\mathcal{U} = \emptyset$ **then**
9:         $\mathcal{U} \leftarrow \{\arg\max_{i:x_i=[\texttt{MASK}]} c_i\}$ {Ensure progress}
10:    **end if**
11:    For all $i \in \mathcal{U}$:   $x_i \leftarrow \arg\max_{v \in \mathcal{V}} p_\theta(v \mid x, q, D)$

12:    *// — Retrieve: refresh evidence —*
13:    Construct proxy $\tilde{x}$ by filling masks with $c_i \geq \tau_q$ via argmax (Equation 3)
14:    $s \leftarrow q \,\|\, \text{Detokenize}(\tilde{x})$ {Form retrieval query}
15:    $D \leftarrow \mathcal{R}(s, K)$ {Retrieve new evidence}
16: **end while**
17: **return** $x$

---

### 4.3. Confidence-Based Unmasking

Because SARDI refreshes evidence at every step, the order in which tokens are committed directly shapes downstream retrieval. We adopt threshold-based unmasking (Wu et al., 2026), which reveals all positions whose confidence exceeds the commit threshold $\tau_c$:

$$\mathcal{U}^t = \{i \mid x_i^t = [\texttt{MASK}] \ \wedge \ c_i \geq \tau_c\}, \quad (6)$$

where $c_i = \max_{v \in \mathcal{V}} p_\theta(v \mid x^t, q, D^t)$. Each unmasked position commits to its argmax:

$$x_i^{t-1} \leftarrow \arg\max_{v \in \mathcal{V}} p_\theta(v \mid x^t, q, D^t). \quad (7)$$

If no position exceeds $\tau_c$, the single most confident masked position is unmasked to guarantee progress.

This creates a natural curriculum: high-confidence tokens (often text spans that can already be grounded in the current evidence) commit first and inform the next retrieval, while uncertain spans wait for refined evidence. The commit threshold $\tau_c$ therefore controls both decoding parallelism and the granularity of retrieval refinement, offering a single knob to trade accuracy against throughput (Figure 3). Algorithm 1 gives the full procedure.

*Table 1.* Main results on multi-hop QA benchmarks (Exact Match $\times 100$). $\pm$ denotes bootstrap standard deviation; Time is mean wall-clock seconds per example. CofCA and SynthWorlds-RM use *counterfactual* corpora (made-up facts) to prevent data-leakage. Search-R1 is grayed as it requires additional RL training (see Section 5.3).

| Method | 2WikiMultihopQA | | HotpotQA | | CofCA | | MuSiQue | | SynthWorlds-RM | |
|---|---|---|---|---|---|---|---|---|---|---|
| | EM | Time | EM | Time | EM | Time | EM | Time | EM | Time |
| *Training-free; autoregressive* | | | | | | | | | | |
| AR W/ RET@STATIC | $44.5_{\pm 0.6}$ | 0.74 | $40.3_{\pm 0.8}$ | 0.66 | $41.0_{\pm 1.6}$ | 0.76 | $12.7_{\pm 0.7}$ | 0.69 | $16.5_{\pm 1.1}$ | 0.81 |
| AR W/ RET@10 | $53.6_{\pm 0.6}$ | 0.80 | $45.1_{\pm 0.8}$ | 0.75 | $39.6_{\pm 1.6}$ | 0.83 | $17.7_{\pm 0.8}$ | 0.74 | $19.3_{\pm 1.1}$ | 0.95 |
| AR W/ RET@1 | $58.8_{\pm 0.6}$ | 1.26 | $47.4_{\pm 0.8}$ | 1.40 | $41.2_{\pm 1.6}$ | 1.37 | $19.8_{\pm 0.8}$ | 1.23 | $20.4_{\pm 1.1}$ | 2.08 |
| AR W/ RET@ADAPTIVE | $46.6_{\pm 0.6}$ | 1.47 | $37.3_{\pm 0.8}$ | 1.34 | $32.4_{\pm 1.6}$ | 1.49 | $17.0_{\pm 0.8}$ | 1.42 | $15.7_{\pm 1.1}$ | 1.71 |
| AdaptiveRAG (Jeong et al., 2024) | $34.1_{\pm 0.6}$ | 5.37 | $37.1_{\pm 0.8}$ | 4.07 | $38.4_{\pm 1.6}$ | 2.88 | $14.9_{\pm 0.7}$ | 7.38 | $13.2_{\pm 1.0}$ | 8.94 |
| ReAct (Yao et al., 2022) | $42.7_{\pm 0.6}$ | 2.15 | $40.1_{\pm 0.8}$ | 2.07 | $42.9_{\pm 1.6}$ | 2.02 | $20.9_{\pm 0.8}$ | 2.32 | $22.2_{\pm 1.2}$ | 3.01 |
| *Training-free; diffusion* | | | | | | | | | | |
| DLM W/ RET@STATIC $\tau_c$=0.9 | $43.7_{\pm 0.6}$ | 0.46 | $39.9_{\pm 0.8}$ | 0.55 | $43.4_{\pm 1.6}$ | 0.79 | $11.1_{\pm 0.7}$ | 1.01 | $14.4_{\pm 1.0}$ | 1.49 |
| DLM W/ SARDI (Ours) $\tau_c$=0.9 | $57.8_{\pm 0.6}$ | 0.39 | $48.5_{\pm 0.8}$ | 0.64 | $45.3_{\pm 1.6}$ | 0.75 | $20.5_{\pm 0.8}$ | 0.88 | $21.1_{\pm 1.2}$ | 1.29 |
| DLM W/ SARDI (Ours) $\tau_c$=0.95 | $59.1_{\pm 0.6}$ | 0.56 | $48.7_{\pm 0.8}$ | 0.90 | $44.9_{\pm 1.6}$ | 1.09 | $20.6_{\pm 0.8}$ | 1.19 | $21.7_{\pm 1.2}$ | 1.78 |
| *RL-trained; autoregressive* | | | | | | | | | | |
| AR (Search-R1) (Jin et al., 2025) | $52.4_{\pm 0.6}$ | 3.36 | $50.3_{\pm 0.8}$ | 2.83 | $44.4_{\pm 1.6}$ | 3.00 | $26.4_{\pm 0.9}$ | 3.14 | $26.9_{\pm 1.3}$ | 3.74 |

## 5. Experiments

### 5.1. Experimental Setup

**Datasets and benchmarks.** We evaluate SARDI on five multi-hop QA benchmarks: 2WikiMultiHopQA (Ho et al., 2020), HotpotQA (Yang et al., 2018), MuSiQue (Trivedi et al., 2022), CofCA (Wu et al., 2025), and SynthWorlds-RM (Gu et al., 2025), reporting Exact Match (EM) following Jin et al. (2025). To measure per-query retrieval recall, the corpus must be at the same passage granularity as each benchmark's gold annotations: HotpotQA provides a full Wikipedia corpus at this granularity (Yang et al., 2018), and for the other four we follow prior work (Prabhu & Anand, 2024; Yu et al., 2026) and concatenate all gold and distractor passages into a single corpus. Unless stated otherwise we use the sparse retrieval method BM25 (Robertson et al., 2009), retrieving $K$=7 passages per iteration, and additionally evaluate the *E5-base-v2* dense retriever in Section 5.6. Latency is measured in wall-clock time unbatched on a single NVIDIA B200 GPU.

**RAG and reasoning with DLMs.** SARDI relies on the existence of a reasoning trace in order to surface bridge entities. In our analysis, off-the-shelf instruction-tuned diffusion LMs such as DREAM-7B almost never produce reasoning traces in a RAG setting, even when using few-shot prompting (Table 6, App. A). Trivedi et al. (2023) made an analogous observation for AR models three years ago: "IRCoT relies on the base LM to have a zero or few-shot CoT-generation ability [...] not as common for small LMs (under 20B) [...] smaller LMs will likely increasingly acquire such ability." (Trivedi et al., 2023) Their prediction

proved correct for AR models, and we expect the same for DLMs as they mature.

For now, to evaluate our retrieval mechanism fairly, we elicit the model to output reasoning via light supervised fine-tuning on synthetic chain-of-thought traces from *gpt-4o-mini* (App. A). To ensure a fair comparison, we apply *identical* fine-tuning to DREAM-7B and the AR model Qwen2.5-7B. Crucially, after fine-tuning the two models reach near-identical EM under static question-only retrieval ($K$=7) on 2WikiMultiHopQA: DLM W/ RET@STATIC achieves 43.7% and AR W/ RET@STATIC achieves 44.5% (Table 1). To ensure the gains do not come from memorization, we include two counterfactual benchmarks. CofCA and SynthWorlds-RM use corpora with made-up facts that are absent from both pre-training and our fine-tuning data. Finally, we emphasize that SARDI is not tied to the specific reasoning format induced by the synthetic *gpt-4o-mini* traces; any reasoning style that includes intermediate entities will work.

### 5.2. Baselines

We compare SARDI against retrieval-augmented baselines from both AR and diffusion paradigms. AR baselines use Qwen2.5-7B and diffusion baselines use DREAM-7B, with the same light fine-tuning (Section 5.1) applied to both backbones. The one exception is Search-R1, which we evaluate from the authors' released RL-trained checkpoint (App. C). All methods share identical prompts and evidence formatting (App. B), isolating the retrieval mechanism.

- **AR W/ RET@STATIC**: Single retrieval from $q$; $D$ is fixed throughout generation.

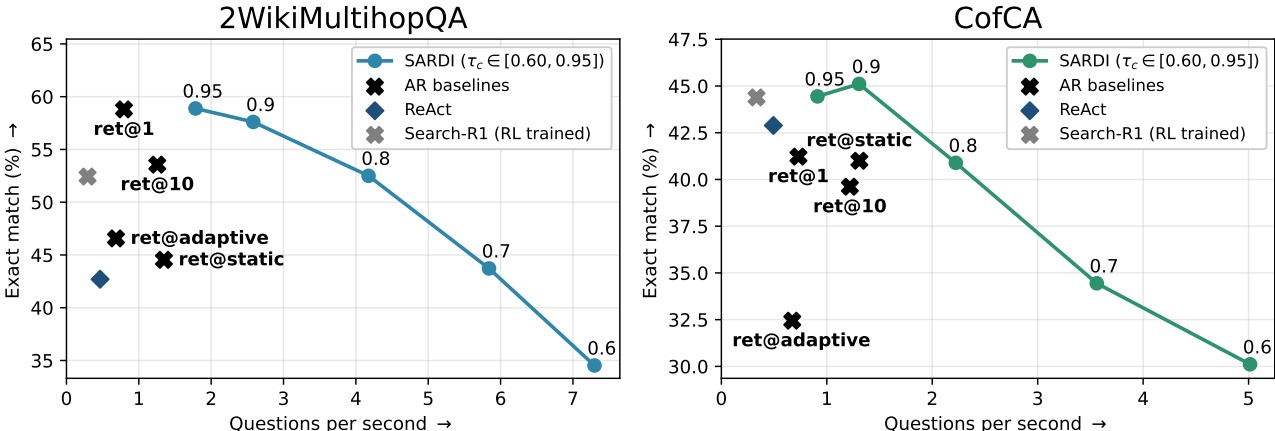

*Figure 3.* Accuracy vs. throughput trade-off on 2WikiMultiHopQA and CofCA, traced by adjusting the commit threshold $\tau_c$. Similar pareto patterns hold on HotpotQA and MuSiQue (Figure 6).

- **AR W/ RET@N**: Retrieval every $N \in \{1, 10\}$ AR tokens.

- **AR W/ RET@ADAPTIVE** (Jiang et al., 2023) (FLARE): Confidence-triggered retrieval using low-confidence spans as both triggers and queries.

- **AdaptiveRAG** (Jeong et al., 2024): AR LM equipped with a query-complexity router that triggers either single-step or iterative retrieval.

- **ReAct** (Yao et al., 2022): A training-free agentic loop with `Retrieve[query]` and `Finish[answer]` actions.

- **DLM W/ RET@STATIC**: The diffusion counterpart of AR W/ RET@STATIC; one retrieval, fixed $D$.

- **DLM W/ SARDI** (this work): DREAM-7B equipped with SARDI.

- **AR (Search-R1)** (Jin et al., 2025): An RL-trained search agent that learns to emit explicit search queries during generation. Included as a strong reference point; discussed in Section 5.3.

### 5.3. SARDI Is Faster and More Accurate

**Accuracy and throughput.** SARDI improves substantially over static diffusion retrieval on every benchmark (Table 1): on 2WikiMultiHopQA it raises EM from 44 to 59, with similar gains on HotpotQA ($40 \rightarrow 49$), CofCA ($43 \rightarrow 45$), and MuSiQue ($11 \rightarrow 21$). It also matches or beats every training-free AR baseline, and does so at much lower latency. We can trade off throughput against accuracy via the commit confidence threshold $\tau_c$: a higher $\tau_c$ commits fewer tokens per denoising step and triggers more retrieval rounds (higher accuracy, lower throughput), while a lower $\tau_c$ does the reverse. Sweeping $\tau_c$ (Figure 3), SARDI

dominates the quality–latency frontier, running up to $8\times$ faster than AR iterative-retrieval baselines at comparable or better accuracy. We analyze the sources of these gains in the following sections.

**Comparison to trained search agents.** Search-R1 (Jin et al., 2025) is an RL-trained search agent that learns to emit explicit search queries during generation. We include it as a strong reference point but emphasize that **SARDI and Search-R1 occupy different design points**: Search-R1 invests in RL-based query generation to maximize accuracy at inference cost, whereas SARDI is a training-free, plug-and-play retrieval mechanism that requires no reward design or policy optimization. Across the five benchmarks SARDI is the strongest training-free method while running $3$–$8\times$ faster than Search-R1 at comparable accuracy on 2Wiki-MultiHopQA, HotpotQA, and CofCA (Table 1). The two approaches are complementary rather than competing, and combining diffusion trajectories with light RL-based query supervision is a natural direction for future work.

### 5.4. Diffusion Trajectories as Lookahead for Retrieval

A central hypothesis of this work is that the uncommitted, low-confidence tokens in the diffusion trajectory are a useful lookahead signal for retrieval. Because a DLM denoises the whole sequence at once, it holds tentative predictions for every position long before they are committed. SARDI feeds these speculative tokens to the retriever, surfacing later-hop evidence early on in the generation process. The following experiments test this hypothesis.

**More lookahead is better.** The query threshold $\tau_q$ sets how confident the model must be in a token before exposing it to the retriever (Equation (3)): $\tau_q = 0$ exposes every position, while $\tau_q = \tau_c$ restricts the query to tokens that are

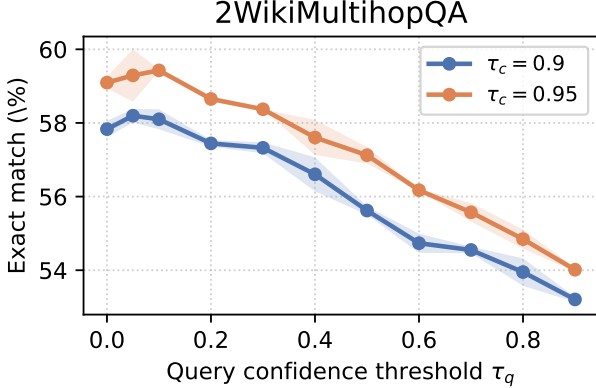

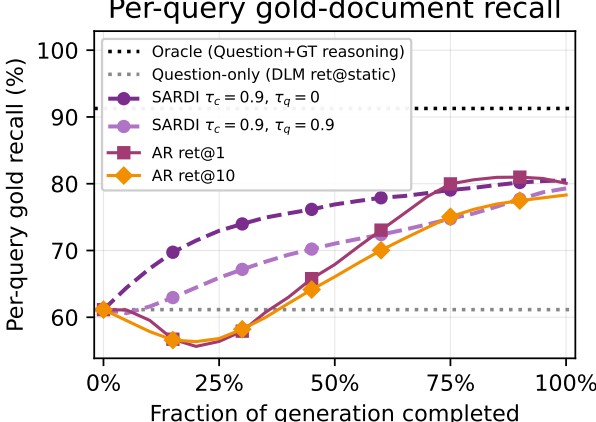

*Figure 4.* Sweep of the query threshold $\tau_q$ on 2WikiMultiHopQA (BM25, $K=7$), at the two commit thresholds $\tau_c \in \{0.9, 0.95\}$. Similar trend holds on HotpotQA, MuSiQue, and SynthWorlds-RM (Table 11).

*Figure 5.* Per-query gold-document recall (fraction of gold passages in the *current* retrieved set) as generation progresses ($K=7$, BM25, 2WikiMQA). We restrict to methods that retrieve at every (or every-$N^{\text{th}}$) token; agentic methods and FLARE issue specific targeted queries by design, so their per-query recall is low even though cumulative recall is comparable (Appendix D, Table 8).

confident enough to be committed. Sweeping $\tau_q$ on 2Wiki-MultiHopQA (Figure 4), EM peaks at the most aggressive setting ($\tau_q \approx 0$) and falls as the query becomes more conservative; similar trends hold on HotpotQA, MuSiQue, and SynthWorlds-RM (Table 11). This directly supports our hypothesis that speculative future tokens can inform retrieval long before they are stable enough to commit to the output.

**Lookahead surfaces evidence earlier.** To further analyze the effect of lookahead, Figure 5 plots per-query document recall (the fraction of gold passages in the current retrieved set) against generation progress. Two horizontal lines provide baselines: the recall of a question-only query (what static retrieval sees) and the recall of a query built from the ground-truth reasoning trace (roughly the best our framework could reach). As expected, early in generation SARDI sits well above the AR baselines (a +19-point recall gain at 25% of generation) and closes much of the gap toward the oracle line. As expected, the more aggressive the lookahead (lower $\tau_q$), the higher the early recall. By the final step SARDI converges to AR ret@1, so its main advantage is that strong evidence arrives *early*, letting it make more progress in the opening steps.

**Gains concentrate on multi-hop reasoning.** Table 2 breaks 2WikiMultiHopQA down by question type: SARDI yields over $2.5\times$ EM gains on *inference* and *compositional* questions (which require identifying bridge entities through multi-step reasoning) and leaves single-hop (*comparison*) questions (where static retrieval already suffices) unchanged.

*Table 2.* Question-type breakdown on 2Wiki (EM %). SARDI benefits most from question types requiring multi-step reasoning.

| Method | 1-hop | Multi-hop | | |
|---|---|---|---|---|
| | comp. | bridge-comp. | compos. | infer. |
| DLM W/ RET@STATIC | 85.6 | 65.3 | 16.6 | 14.0 |
| DLM W/ SARDI(Ours) $\tau_c$=0.95 | $84.6_{-1.0}$ | $71.7_{+6.4}$ | $45.3_{+28.7}$ | $37.5_{+23.5}$ |

### 5.5. RAG Grounding Promotes Parallel Decoding

We argued in Section 1 that retrieved evidence reduces inter-token dependence, making RAG well-suited to parallel decoding. We test this by measuring the conditional mutual information (CMI) between adjacent tokens in ground-truth reasoning traces on 2WikiMultiHopQA:

$$\text{CMI}(x_i; x_{i+1} \mid D) = \\ \mathbb{E}_{x_i}\big[\text{KL}\big(p(x_{i+1} \mid x_i, D) \, \| \, p(x_{i+1} \mid D)\big)\big].$$

We approximate the expectation over $x_i$ by its top-7 values and vary the amount of retrieved evidence in $D$.

Intuitively, CMI is high when several completions are plausible and the model has not settled on one. Multi-token names are the clearest case: until the model commits to *which* entity it is naming, the tokens must agree to stay coherent (*Albert Einstein* or *Isaac Newton*, not *Albert Newton*), so fixing one sharply shifts the distribution over the other. Strong evidence removes this coupling: the model copies the entity straight from the retrieved passage instead of coordinating across positions.

The results in Table 3 confirm this. With all gold documents present, entity pairs show very low dependence (CMI=0.060), i.e., they can be decoded in parallel. As gold

*Table 3.* Conditional mutual information (CMI) between adjacent tokens in DREAM-7B reasoning traces (2WikiMultiHopQA), as gold documents are progressively removed from the context. We separately report entity and non-entity pairs.

|  | All gold | Gold$-1$ | Gold$-2$ | No gold |
|---|---|---|---|---|
| CMI (entity) | 0.060 | 0.219 | 0.396 | 0.588 |
| CMI (non-entity) | 0.136 | 0.214 | 0.247 | 0.264 |

documents are removed, entity-pair CMI rises nearly $10\times$ to 0.588, while non-entity pairs increase from 0.136 to 0.264. Grounding thus removes the inter-token dependence most harmful to parallel decoding: tightly coupled entity spans. This confirms that the RAG setting is well suited to parallel decoding.

### 5.6. Ablations

**Retriever choice.** SARDI is retriever-agnostic. Replacing BM25 with the E5-base-v2 dense retriever, SARDI still outperforms the strongest training-free AR baseline (AR W/ RET@1) and stays competitive with the RL-trained Search-R1 (Table 4). Also see more detailed recall results in Appendix D. Thus, the observed behavior does not depend on the lexical retriever.

*Table 4.* SARDI and strongest baselines with a dense retriever (E5-base-v2). EM ($\times100$), $K=7$.

| Method | 2Wiki | Hotpot | CofCA | MuSiQue | SynthW. |
|---|---|---|---|---|---|
| Search-R1 | 41 | 49 | 44 | 26 | 21 |
| AR W/ RET@1 | 47 | 47 | 41 | 20 | 22 |
| DLM W/ SARDI (Ours) | 50 | 52 | 45 | 22 | 23 |

**Refresh schedule.** SARDI refreshes retrieval at every denoising step, which is wasteful when retrieval is expensive (rerankers, large corpora, tight serving budgets). Table 5 varies the refresh frequency: refreshing every 2 steps costs only 1–2 EM. Moreover, we show in Appendix E that the retrieved set is relatively stable: on average 83–90% of documents persist between consecutive steps. Both findings suggest that the per-step retrieval cost can be substantially amortized – a direction we discuss in the Limitations section.

*Table 5.* SARDI accuracy under different retrieval refresh frequencies. EM ($\times100$); subscripts give the change from refreshing every step.

| Refresh | 2Wiki | HotpotQA | CofCA | MuSiQue |
|---|---|---|---|---|
| every step | 58 | 48 | 45 | 20 |
| every 2 | $56_{-2}$ | $47_{-1}$ | 45 | 20 |
| every 4 | $52_{-6}$ | $46_{-2}$ | 45 | $18_{-2}$ |

## 6. Limitations

While SARDI is training-free in principle, the tested diffusion language models do not yet reliably produce reasoning-traces through prompting alone. This is analogous to what Trivedi et al. (2023) observed for early medium-scale autoregressive models, which has since been resolved as AR models matured; we expect the same to happen for DLMs, removing the need for the fine-tuning step. Additionally, our method refreshes retrieval at every denoising step, which can incur document encoding overhead under naive implementations. Because a large portion of the retrieved documents persists between consecutive steps, integrating Fast-dLLM-style block KV caching is a natural direction for future work. Extending self-augmenting retrieval to latent diffusion language models is another promising direction.

## 7. Conclusion

We introduced **Self-Augmenting Retrieval (SARDI)**, a training-free, dynamic retrieval framework that exploits two structural opportunities that diffusion language models open up for retrieval-augmented generation. First, the denoising trajectory exposes tentative predictions for the entire response at every step, surfacing salient entities early and turning them into a lookahead signal for retrieval. Second, we have shown that grounding generation in retrieved evidence sharply reduces inter-token dependence, making RAG a regime especially well suited to parallel decoding. SARDI leverages both: it interleaves retrieval with denoising, refining the query as the response takes shape. On five multi-hop QA benchmarks, SARDI substantially improves over static retrieval, matches or beats every training-free AR baseline, and runs up to $8\times$ faster. More broadly, denoising trajectories offer a natural signal for dynamic retrieval, and we expect SARDI to strengthen as diffusion language models mature.

## Acknowledgements

We thank Alexander Kreibich and Lucas He for productive discussions that helped motivate this work. JL is supported by a Google PhD Fellowship, LZ by a LinkedIn PhD Fellowship, and DG by an Empire AI Postdoctoral Fellowship. We are grateful to NVIDIA for access to the DGX Station compute platform through their Early Access Program. This work was further supported by the National Science Foundation (NSF) under grants OAC-2118310 and 2530143, through the AI Research Institutes program (Award No. DMR-2433348), by funding from NewYork-Presbyterian for the NYP-Cornell Cardiovascular AI Collaboration, and by arXiv. Finally, we thank the anonymous reviewers, whose feedback strengthened the paper by motivating the analyses of early-generation recall and mutual information.

## Impact Statement

This work studies retrieval-augmented generation for diffusion language models. Improving evidence grounding can reduce hallucinations in knowledge-intensive settings, but the same retrieval mechanisms could produce more convincing misinformation if paired with untrusted corpora. We recommend deploying SARDI with curated sources and provenance tracking, such as citing retrieved passages alongside generated answers. All experiments use publicly available QA benchmarks and open retrieval corpora.

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

## A. Fine-Tuning Analysis

*Table 6.* Output-mode breakdown for base and fine-tuned DREAM-7B and Qwen2.5-7B on 2WikiMultiHopQA (static $K$=7 RAG, $n$=500).

| | DREAM-7B | Qwen2.5-7B | DREAM-7B (SFT) | Qwen2.5-7B (SFT) |
|---|---|---|---|---|
| Correct (with reasoning trace) | 1% | 32% | 44% | 44% |
| Correct (no reasoning trace) | 27% | 0% | 0% | 0% |
| Wrong | 72% | 68% | 56% | 56% |

To motivate the fine-tuning stage and isolate its effect from the retrieval mechanism, we present an output analysis of base and fine-tuned models on 500 examples from 2WikiMultiHopQA under static $K$=7 retrieval. Table 6 reports the proportion of outputs in three categories: correct with a structured reasoning trace, correct without a reasoning trace, and wrong (which folds in format errors such as missing the ### answer marker).

Two observations are worth highlighting:

1. **Base DREAM-7B cannot produce structured reasoning from prompting alone.** Only 1% of base DREAM-7B outputs include any reasoning trace at all—most simply emit a guess. The same-sized AR Qwen2.5-7B reasons in 32% of cases. SARDI relies on intermediate states surfacing salient entities, so this failure mode makes SARDI inapplicable without fine-tuning.

2. **Fine-tuning equalizes capability.** On this $n$=500 subset both models land at 44% correct-with-reasoning; on the full test set they sit within 1 EM of each other (DLM W/ RET@STATIC=43.7%, AR W/ RET@STATIC=44.5%; Table 1). Any subsequent gap between SARDI and matched-fine-tuning AR baselines is therefore attributable to the retrieval mechanism rather than the underlying model.

This is analogous to what Trivedi et al. (2023) observed for early small AR models: "IRCoT relies on the base LM to have a zero or few-shot CoT-generation ability. While this is commonly available in large LMs (over 100B), it's not as common for small LMs (under 20B)...smaller LMs will likely increasingly acquire such ability." That prediction proved correct for AR models; we expect the same for DLMs.

### A.1. Fine-Tuning Data

We construct supervised fine-tuning data from the training splits of 2WikiMultiHopQA (Ho et al., 2020) and HotpotQA (Yang et al., 2018), augmented with synthetically generated chain-of-thought reasoning traces.

**Synthetic reasoning traces.** For each training example, we prompt *gpt-4o-mini* with the question $q$, gold answer $a$, and supporting documents to generate a step-by-step reasoning trace that (i) identifies intermediate bridge entities, (ii) references specific facts from the supporting documents, and (iii) concludes with the final answer in the format ### [answer].

**Example.** The following illustrates a training example from 2WikiMultiHopQA:

```
Question: Where was the director of film Ronnie Rocket born?

Reasoning (synthetically generated):
Step 1: Ronnie Rocket is directed by David Lynch.
Step 2: David Lynch was born in Missoula, Montana.

### Montana
```

**Training context.** During training, we present each training example together with its gold documents, reinforcing Dream-7B to generate reasoning traces when solving RAG problems.

## A.2. Fine-Tuning Configuration

We fine-tune DREAM-7B using fully sharded data parallel (FSDP) on $2\times$ NVIDIA B200 GPUs. Table 7 summarizes the hyperparameters. To ensure a fair comparison, we apply the same configuration and training data to all autoregressive baselines (Qwen2.5-7B), isolating differences attributable to the generation paradigm.

*Table 7.* Fine-tuning hyperparameters.

| Hyperparameter | Value |
|---|---|
| Base model | DREAM-7B |
| Training epochs | 3 |
| Learning rate | $2 \times 10^{-6}$ |
| Global batch size | 256 |
| Micro batch size per GPU | 16 |
| Maximum sequence length | 2048 tokens |
| Optimizer | AdamW |
| Hardware | $2\times$ NVIDIA B200 GPUs |

## B. Prompt Templates

### B.1. RAG Input Format

All models receive input in the following format:

```
Use ONLY the provided facts to answer the question.
Think step-by-step, then provide the final answer after the "###" marker.

Question:
{question}

Facts:
{facts}
```

where {question} is the input question and {facts} contains the concatenated retrieved passages.

### B.2. Expected Output Format

Models are trained to produce outputs in the following format:

```
Step 1: [First reasoning step grounded in documents]
Step 2: [Second reasoning step]
...
### [final answer]
```

This format serves two purposes: it enables extraction of intermediate entities from partial reasoning traces for self-augmenting retrieval, and it provides a consistent extraction point (###) for evaluation.

## C. Additional Experimental Details

**Evaluation protocol.**   We extract the final answer by parsing text following the ### marker. Exact Match (EM) is computed after normalizing both predicted and gold answers and checking for string equality.

**Retrieval configuration.**   We use BM25 (Robertson et al., 2009) for all experiments unless otherwise specified. For the dense-retriever results in Section 5.6 we use E5-base-v2 (Wang et al., 2022). For SARDI, retrieval is performed at every denoising iteration using the concatenation of the original question and the current intermediate response as the query, with $K=7$ passages per iteration unless otherwise specified.

**Search-R1 setup.** The four AR baselines (AR W/ RET@STATIC, AR W/ RET@10, AR W/ RET@1, AR W/ RET@ADAPTIVE) and the training-free agentic baselines (AdaptiveRAG, ReAct) all share the same Qwen2.5-7B backbone, supervised-fine-tuned on the combined training sets of 2WikiMultiHopQA and HotpotQA (Appendix A); any accuracy gap with SARDI therefore isolates the retrieval mechanism. Search-R1 is the only baseline that does *not* share this backbone: we evaluate the authors' publicly released checkpoint, which they trained via PPO on HotpotQA using a multi-stage reward (Jin et al., 2025). Reproducing this RL training pipeline was not feasible within our compute budget, so we accept this additional confound and treat Search-R1 as an RL-trained reference point rather than a controlled comparison (hence the gray styling in Table 1).

## D. Per-Method Recall: Full Tables

Figure 5 in the main text reports gold-document recall over generation progress for SARDI and the AR retrieval baselines that fire at every (or every-$N^{\text{th}}$) token. For completeness, this section reports the full per-method numbers, including the agentic methods (AdaptiveRAG, ReAct, Search-R1) and AR speculative-lookahead (FLARE).

We report two complementary recall metrics:

- **Per-query final recall** (Table 8): fraction of gold documents present in the retrieved set $D$ at the *final* retrieval step, averaged over questions. This is the metric most directly tied to answer accuracy on a single forward pass.

- **Cumulative recall** (Table 9): fraction of gold documents that have appeared in $D$ at *any* point during generation, i.e., the union over all retrieval steps. This view is fairer to methods that issue few but targeted queries.

**Caveat: cumulative recall depends on retrieval volume.** A method that issues many queries (e.g., SARDI, AR ret@1) trivially accumulates more unique documents than one that issues few targeted queries (e.g., Search-R1), and cumulative recall therefore is *not* directly comparable across methods without controlling for total documents touched. Table 9 accordingly reports the average number of unique documents retrieved per question as a subscript next to each cumulative-recall value, so the reader can normalize.

*Table 8.* Per-query gold-document recall at the final retrieval step ($\times 100$). Methods that issue specific targeted queries (AdaptiveRAG, ReAct, AR ret@adaptive/FLARE, Search-R1) retrieve few documents per query, so per-query recall is low by design even when their cumulative coverage (Table 9) is comparable. SARDI matches AR ret@1 across $K$ and retrievers.

| | **2Wiki** | | | | | | | | **MuSiQue** | | | | | | | |
| | *BM25* | | | | *Dense* | | | | *BM25* | | | | *Dense* | | | |
| **Method** | $K$=3 | $K$=7 | $K$=10 | $K$=15 | $K$=3 | $K$=7 | $K$=10 | $K$=15 | $K$=3 | $K$=7 | $K$=10 | $K$=15 | $K$=3 | $K$=7 | $K$=10 | $K$=15 |
|---|---|---|---|---|---|---|---|---|---|---|---|---|---|---|---|---|
| AdaptiveRAG | 5 | 5 | 4 | 3 | – | 5 | 4 | 3 | 9 | 10 | 9 | 7 | – | 9 | – | – |
| ReAct | 38 | 48 | 51 | 54 | 27 | 31 | 31 | 33 | 26 | 34 | 37 | 40 | 27 | 31 | 32 | 35 |
| AR ret@adaptive (FLARE) | 22 | 27 | 29 | 32 | 14 | 17 | 18 | 19 | 14 | 20 | 22 | 25 | 14 | 20 | 21 | 22 |
| AR ret@10 | 65 | 78 | 82 | 86 | 54 | 61 | 62 | 63 | 42 | 53 | 56 | 61 | 38 | 47 | 51 | 54 |
| AR ret@1 | 66 | 80 | 85 | 88 | 54 | 61 | 63 | 64 | 43 | 54 | 58 | 62 | 38 | 48 | 51 | 54 |
| Search-R1 | 42 | 53 | 57 | 62 | 28 | 31 | 33 | 35 | 29 | 38 | 42 | 45 | 28 | 35 | 37 | 41 |
| **SARDI** $\tau_c$=0.9 | 66 | 81 | 83 | 86 | 55 | 63 | 62 | 63 | 43 | 55 | 56 | 60 | 39 | 50 | 53 | 55 |
| **SARDI** $\tau_c$=0.95 | 67 | 81 | 84 | 88 | 56 | 63 | 64 | 65 | 44 | 56 | 59 | 64 | 40 | 50 | 53 | 56 |

## E. Document Persistence Across Refresh Steps

Table 5 relies on the observation that retrieved documents change only gradually between consecutive denoising steps, so document-level KV-cache reuse is feasible. We measure the average overlap between consecutive retrieved sets as $|D^t \cap D^{t-1}|/|D^{t-1}|$:

| **Dataset** | **% docs retained per step** |
|---|---|
| 2WikiMultiHopQA | 88% |
| HotpotQA | 83% |
| CofCA | 89% |
| MuSiQue | 84% |
| SynthWorlds-RM | 90% |

*Table 9.* Cumulative gold-document recall ($\times100$, main number) and average number of unique documents retrieved per question ($_{subscript}$). Cumulative recall measures the union of all gold documents surfaced at any point during generation, but is *not directly comparable across methods that retrieve different total document counts*: a method that issues many queries (e.g., SARDI, AR ret@1) trivially accumulates more docs than one that issues few (e.g., Search-R1). The subscripts let the reader normalize. Even controlling for total docs, SARDI remains competitive with the strongest baselines.

| Method | 2Wiki | | | | | | | | MuSiQue | | | | | | | |
| | *BM25* | | | | *Dense* | | | | *BM25* | | | | *Dense* | | | |
| | $K=3$ | $K=7$ | $K=10$ | $K=15$ | $K=3$ | $K=7$ | $K=10$ | $K=15$ | $K=3$ | $K=7$ | $K=10$ | $K=15$ | $K=3$ | $K=7$ | $K=10$ | $K=15$ |
|---|---|---|---|---|---|---|---|---|---|---|---|---|---|---|---|---|
| AdaptiveRAG | $71_6$ | $79_{13}$ | $77_{14}$ | $62_{15}$ | – | – | – | – | $48_7$ | $56_{13}$ | $55_{14}$ | $48_{15}$ | – | – | – | – |
| ReAct | $73_7$ | $81_{15}$ | $82_{21}$ | $83_{31}$ | $57_7$ | $57_{15}$ | $57_{21}$ | $58_{31}$ | $50_6$ | $60_{15}$ | $63_{22}$ | $65_{33}$ | $54_7$ | $60_{16}$ | $62_{22}$ | $64_{33}$ |
| AR ret@adaptive (FLARE) | $73_8$ | $82_{20}$ | $84_{29}$ | $87_{43}$ | $63_9$ | $66_{22}$ | $67_{32}$ | $68_{49}$ | $51_9$ | $62_{22}$ | $65_{31}$ | $69_{48}$ | $52_9$ | $60_{22}$ | $64_{32}$ | $65_{48}$ |
| AR ret@10 | $71_6$ | $83_{15}$ | $86_{22}$ | $89_{34}$ | $61_6$ | $66_{15}$ | $67_{22}$ | $68_{33}$ | $49_6$ | $60_{16}$ | $64_{23}$ | $68_{34}$ | $46_6$ | $55_{15}$ | $58_{22}$ | $62_{32}$ |
| AR ret@1 | $73_7$ | $85_{20}$ | $89_{29}$ | $91_{46}$ | $64_9$ | $68_{23}$ | $69_{33}$ | $70_{50}$ | $52_8$ | $63_{21}$ | $67_{30}$ | $71_{45}$ | $50_9$ | $59_{22}$ | $62_{31}$ | $65_{46}$ |
| Search-R1 | $74_6$ | $85_{16}$ | $88_{25}$ | $90_{34}$ | $60_7$ | $63_{17}$ | $63_{25}$ | $63_{37}$ | $57_7$ | $65_{16}$ | $68_{23}$ | – | $59_7$ | $65_{17}$ | $67_{23}$ | $70_{34}$ |
| SARDI $\tau_c$=0.9 | $70_5$ | $83_{14}$ | $85_{20}$ | $88_{31}$ | $61_6$ | $68_{17}$ | $66_{24}$ | $67_{37}$ | $47_7$ | $63_{19}$ | $61_{28}$ | $65_{44}$ | $47_7$ | $58_{19}$ | $59_{28}$ | $61_{42}$ |
| SARDI $\tau_c$=0.95 | $71_5$ | $83_{14}$ | – | $90_{33}$ | $63_7$ | $68_{18}$ | $70_{26}$ | $71_{40}$ | $52_7$ | $64_{20}$ | $68_{30}$ | $73_{46}$ | $49_8$ | $59_{20}$ | $62_{29}$ | $66_{44}$ |

Since 83–90% of documents persist between consecutive steps, only a small fraction needs recomputation, making document-level KV caching similar to Fast-dLLM (Wu et al., 2026) an attractive direction for combining SARDI with diffusion-decoding-acceleration techniques.

## F. Accuracy vs. Throughput: Full Datasets

Figure 3 in the main text shows the Pareto frontier on 2WikiMultiHopQA and CofCA. Figure 6 reports the same plot across all four benchmarks. SARDI dominates the autoregressive iterative-retrieval baselines on every dataset and matches Search-R1 at 3–8× lower latency.

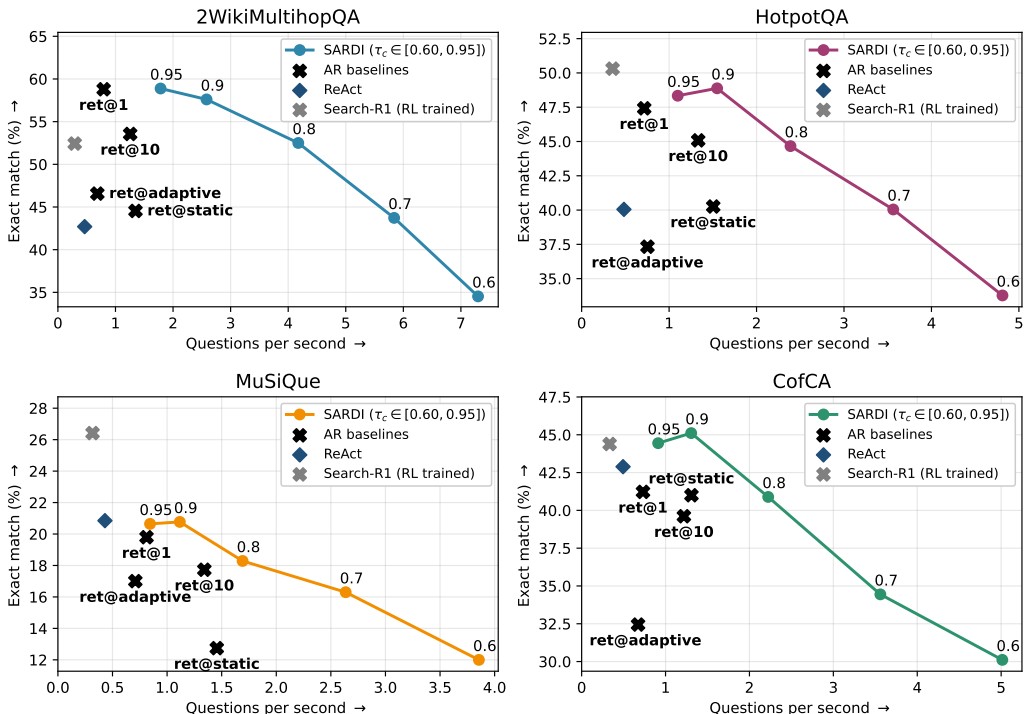

*Figure 6.* Accuracy vs. throughput trade-off across all four benchmarks, companion to Figure 3.

*Table 10.* Threshold-based unmasking matches fixed-step accuracy at 2–3× the speed. EM (×100) and time per example (s).

|  | 2Wiki EM / T | HotpotQA EM / T | CofCA EM / T | MuSiQue EM / T |
|---|---|---|---|---|
| 50 fixed steps | 59 / 1.33 | 47 / 1.46 | 43 / 1.48 | 20 / 1.42 |
| $\tau_c$=0.9 | $58_{-1}$ / $0.39_{-0.94}$ | $48_{+1}$ / $0.64_{-0.82}$ | $45_{+2}$ / $0.75_{-0.73}$ | $20$ / $0.88_{-0.54}$ |
| $\tau_c$=0.95 | $59$ / $0.56_{-0.77}$ | $49_{+2}$ / $0.90_{-0.56}$ | $45_{+2}$ / $1.09_{-0.39}$ | $21_{+1}$ / $1.19_{-0.23}$ |

*Table 11.* Query-threshold sweep, full numbers. EM (%) on the four datasets with $\tau_q$ sweeps, at commit thresholds $\tau_c \in \{0.9, 0.95\}$.

| Dataset | $\tau_c$ | $\tau_q$=0 | $\tau_q$=0.1 | $\tau_q$=0.2 | $\tau_q$=0.3 | $\tau_q$=0.4 | $\tau_q$=0.5 | $\tau_q$=0.6 | $\tau_q$=0.7 | $\tau_q$=0.8 | $\tau_q$=0.9 |
|---|---|---|---|---|---|---|---|---|---|---|---|
| 2WikiMultihopQA | 0.9 | 57.8 | 58.1 | 57.4 | 57.3 | 56.6 | 55.6 | 54.7 | 54.5 | 54.0 | 53.2 |
|  | 0.95 | 59.1 | 59.4 | 58.7 | 58.4 | 57.6 | 57.1 | 56.2 | 55.6 | 54.8 | 54.0 |
| HotpotQA | 0.9 | 48.3 | 48.1 | 47.6 | 46.4 | 45.9 | 46.0 | 44.8 | 44.7 | 44.4 | 44.1 |
|  | 0.95 | 48.7 | 48.4 | 47.8 | 47.0 | 46.3 | 45.6 | 45.3 | 44.9 | 44.4 | 44.1 |
| MuSiQue | 0.9 | 20.5 | 20.6 | 19.7 | 19.2 | 18.2 | 16.7 | 16.3 | 15.7 | 15.7 | 15.2 |
|  | 0.95 | 20.6 | 20.6 | 19.5 | 18.9 | 17.9 | 17.1 | 16.3 | 16.3 | 15.6 | 14.7 |
| SynthWorlds | 0.9 | 21.1 | 21.4 | 21.1 | 19.9 | 17.9 | 17.5 | 17.9 | 17.0 | 16.8 | 17.2 |
|  | 0.95 | 21.7 | 21.5 | 21.3 | 19.8 | 19.5 | 18.2 | 17.5 | 17.5 | 17.0 | 16.9 |

## G. Threshold-Based vs. Fixed-Step Decoding

Table 10 compares fixed-step decoding ($T$=50) against threshold-based unmasking at the two commit thresholds used in the main paper, $\tau_c$=0.9 and $\tau_c$=0.95. Threshold decoding matches fixed-step accuracy at up to 3× the speed.

## H. Query-Threshold Sweep: Full Numbers

Figure 4 in the main text plots the $\tau_q$ sweep on 2WikiMultiHopQA. Table 11 reports full EM numbers across all four benchmarks for which $\tau_q$ sweeps were run, at both commit thresholds $\tau_c \in \{0.9, 0.95\}$ and ten coarse $\tau_q$ values spanning the spectrum from aggressive lookahead ($\tau_q$=0, the default SARDI setting) to the conservative endpoint ($\tau_q$=0.9, querying only on near-committed tokens). The pattern observed on 2WikiMultiHopQA reproduces on HotpotQA, MuSiQue, and SynthWorlds-RM: peak accuracy lands at $\tau_q \in [0, 0.1]$ on every (dataset, $\tau_c$) cell, and EM generally declines as the query threshold tightens. The conservative endpoint loses 4–6 EM relative to the default, consistent with our hypothesis that retrieval benefits from speculative future tokens that are not yet stable enough to commit.

## I. Qualitative Example: Threshold-Based Unmasking

A central hypothesis of this work is that RAG exhibits a structure uniquely amenable to parallel decoding: when retrieved evidence is sufficiently informative, many output tokens become conditionally independent given the evidence and can be committed simultaneously. Threshold-based unmasking (Equation (6)) exploits this by committing all high-confidence tokens at once rather than artificially limiting parallelism. The following example illustrates this behavior for $\tau_c$=0.8, requiring only 2 steps:

*Where was the place of death of the director of Fight Of The Tertia?*

$x_2 = $ "[M][M][M][M][M][M][M][M][M][M][M][M][M]"

$D_2 = $ "Fight of the Tertia is a 1952 West German family film directed by Erik Ode […]"

$x_1 = $ "(1) Fight of the Tertia is directed by Erik Ode. (2) Erik Ode died in [M][M][M]"

$D_1 = $ "Erik Ode (born Fritz Erik Signy Odemar, 6 November 1910 in Berlin, died 19 July 1983 in Kreuth-Weißach) was a […]"

$x_0 = $ "(1) Fight of the Tertia is directed by Erik Ode. (2) Erik Ode died in Kreuth. ### Kreuth"

