# OpenReview forum: "Self-Augmenting Retrieval for Diffusion Language Models"
_ICML.cc/2026/Conference — ICML 2026 regular_

### Official Review · Reviewer_DKzj · 2026-03-09

**Soundness:** 2
**Presentation:** 3
**Significance:** 2
**Originality:** 3
**Overall Recommendation:** 4
**Confidence:** 4

**Summary:**

This paper studies retrieval-augmented generation with diffusion language models on multi-hop QA. The authors propose SARDI, an inference-time method that refreshes retrieval at each denoising step using the question plus the model’s intermediate partial response. The main idea is that intermediate diffusion states expose bridge entities early, which helps retrieve later-hop evidence that static retrieval misses. Experiments on five QA benchmarks show better exact match than static diffusion retrieval, while also reporting 2 to 6 times higher throughput than autoregressive methods at similar accuracy.

**Compliance With Llm Reviewing Policy:**

Affirmed.

**Final Justification:**

The rebuttal addressed my main concerns, and I hope to see the new ablations and analyses in the final revision.

**Key Questions For Authors:**

All concerns have been stated in the previous section.

**Limitations:**

yes

**Strengths And Weaknesses:**

Strengths:
1. The paper has a clear core idea. It uses intermediate diffusion states as evolving retrieval queries. That is simple, intuitive, and easy to understand. The figures also help a lot in better understanding the method.
2. The empirical gains are meaningful in the paper’s chosen setup. SARDI beats static diffusion retrieval on all five datasets. The paper also compares against auto-regressive baselines.
3. One one the most informative and interesting sections to me is the quality-speed tradeoff, which is a real strength. The throughput plots suggest diffusion-based RAG can be much faster than AR systems while staying competitive in accuracy.

Weaknesses:
1. The biggest issue to me is the "training-free" framing across the paper. SARDI itself is inference-time only, but the paper admits the base diffusion models need extra fine-tuning to become RAG-capable. That weakens the practical claim a lot. Moreover, the fine-tuning of the model not only improves the formatting capabilities of the base model, but also makes it tuned for open question answering. While the comparison between AR and DLM models is valid, both AR and DLM models are fine-tuned on synthetic chain-of-thought traces generated by GPT-4o-mini. That means the reported gains are not purely about the retrieval strategy. They also depend on this added supervision. The authors should add DLM and AR results without fine-tuning for completeness and provide qualitative analyses of failure cases in the appendix.
2. The retrieval setup is limited. Everything uses BM25. The paper should show results with at least one dense retriever and more (smaller and larger) DLMs. Analyses on only one model will not prove the effectiveness of the method, especially when already an additional training was required.
3. "Search-R1 achieves slightly higher accuracy". This is not fairly true. On 3 datasets (CofCA, MuSQue, and SynthWorlds), the Search-R1's performance is quite high compared to SARDI (~40% to ~70% improvement!).  The paper leans on speed and engineering simplicity, which is fair, but that also means SARDI is not the strongest system by quality.
4. Some claims are too broad. The paper suggests diffusion models are particularly well-suited for RAG because many tokens become conditionally independent given evidence. That sounds plausible and intuitive, but the paper does not really prove it. It is still more a hypothesis than a demonstrated principle. Or if already proved by someone (I may have missed this), please cite.
5. SARDI refreshes retrieval every iteration. That is brute force. The authors admit this may be excessive. Also, it is mentioned that in the experiments, the retriever retrieves 7 passages per iteration. (1) Why 7? Provide ablation on this. (2) The authors should report the final number of passages (on average) retrieved to answer a question by different DLM and AR experiments, to better show that the comparison is fair.
6. Other plots for other datasets (in Fig 3 and Fig 4) should be presented in the appendix. Are the trends similar in other cases?
7. Citations should be fixed. Some mistakes as examples:
 - Selfrag: Learning to retrieve, generate, and critique through self-reflection --> ICLR 2024
 - Constructing a multi-hop qa dataset for comprehensive evaluation of reasoning steps --> COLING 2020
 - Search-R1: Training LLMs to Reason and Leverage Search Engines with Reinforcement Learning --> COLM 2025

In general, this is a solid paper with a clean idea. However, the "training-free" messaging is overstated, and the evaluation scope is too narrow to fully justify the broad claims (more than 1 model, more than 1 retriever, some plots seem like being cherry-picked; similar evaluations for other studied datasets should be provided in the appendix, qualitative analysis, etc.).

---

> ### Author Rebuttal · Authors · 2026-03-31
>
> We sincerely appreciate the reviewer's detailed and very constructive feedback.
>
> ### Finetuning and "training-free" framing (W1)
>
> We agree this framing needs to be more precise, and we thank the reviewer for pushing on it. SARDI *itself* is an inference-time algorithm that requires no additional training, learned controllers, or modifications to the diffusion objective. However, the base diffusion language model Dream7B requires SFT to become RAG-capable. Even with few-shot prompts, Dream7B does not consistently output reasoning traces (see below). Since SARDI relies on salient entities in the reasoning trace, it cannot be applied to the Dream7B base model.
>
> | | Correct (w/ reasoning) | Correct (no reasoning) | Wrong |
> |---|---|---|---|
> | Dream7B | 1% | 27% | 72% |
> | Qwen2.5-7B | 32% | 0% | 68% |
> | Dream7B (SFT) | 44% | 0% | 56% |
> | Qwen2.5-7B (SFT) | 44% | 0% | 56% |
>
> Trivedi et al. (2023, ACL) made a directly analogous observation for AR models three years ago:
>
> > *"IRCoT relies on the base LM to have a zero or few-shot CoT-generation ability [...] not as common for small LMs (under 20B) [...] smaller LMs will likely increasingly acquire such ability."*
>
> Their prediction proved correct for AR models; we expect the same for DLMs as they mature.
>
> For now, SFT has two effects: improving document understanding and inducing structured reasoning traces. We control for the first by applying identical SFT to both AR and DLM. **After SFT both reach identical 44% static RAG accuracy**, ensuring a fair comparison. The second role (reasoning format) is not fundamental to the method: SARDI only requires that salient entities appear in intermediate text, which we expect can be achieved via few-shot prompting as models improve. This suggests that the comparison does not fundamentally rely on SFT, but on the retrieval mechanism itself.
>
> ### Baselines and Search-R1 (W3)
>
> We add several RAG baselines (Self-RAG, AdaptiveRAG, ReAct), and SARDI outperforms all of them (see R1 for full results). Search-R1 remains the strongest overall, but SARDI is competitive on 3/5 datasets (within 2 EM on HotpotQA/CofCA, ahead on 2Wiki) while being 3--8× faster. We agree the original "slightly higher" framing was misleading and will revise it. **We emphasize that SARDI is not intended to replace agentic methods like Search-R1, but to offer an engineering-simple, high-performance RAG alternative with an adjustable quality–latency tradeoff.** We thank the reviewer for appreciating the quality-speed tradeoff in Figure 3; we will add similar plots for all datasets in the appendix and observe similar trends there.
>
> ### Conditional independence claims (W4)
>
> We thank the reviewer for raising this concern, which motivated a new experiment that adds a valuable contribution to the paper. To support this empirically, we analyze **conditional mutual information (CMI)**, which measures how coupled two adjacent tokens are: low CMI means the joint distribution is well-approximated by the product of marginals, so tokens can be decoded in parallel. We compute $\text{CMI}(x\_i; x\_{i+1} \mid D) = \mathbb{E}\_{x\_i}[\text{KL}(p(x\_{i+1} \mid x\_i, D) \| p(x\_{i+1} \mid D))]$, approximating the expectation over the top-$k$ logits of $x\_i$, on Dream7B reasoning traces from 2WikiMultiHopQA. We partition by NER into entity and non-entity pairs; roughly half the tokens overlap with named entities.
>
> | | Strong context (all gold) | Gold-1 | Gold-2 | Weak context (no gold) |
> |---|---|---|---|---|
> | CMI (entity) | 0.06 | 0.22 | 0.4 | 0.59 |
> | CMI (non-entity) | 0.12 | 0.2 | 0.26 | 0.29 |
>
> Entity CMI rises 10x as the context quality degrades. This directly supports the conditional independence hypothesis and shows why RAG is particularly suited to parallel decoding.
>
> ### Retrieval ablations (W2, W5)
>
> See our response to Reviewer 1 for full retrieval tables: K=7 provides a good accuracy-latency tradeoff; SARDI works with both BM25 and dense retrievers; SARDI achieves high cumulative recall with moderate total documents compared to baselines, showing the gain comes from query quality rather than retrieval volume.
>
> **Retrieval frequency (W5).** We vary how often retrieval is refreshed during denoising:
>
> | Ret. schedule | 2Wiki EM | HotpotQA EM | CofCA EM | MuSiQue EM |
> |---|---|---|---|---|
> | Every step | 58 | 48 | 46 | 20 |
> | Every 2 steps | 56 | 47 | 45 | 20 |
> | Every 4 steps | 52 | 46 | 45 | 18 |
>
> This shows that frequent retrieval is beneficial but not strictly necessary, suggesting more efficient adaptive schedules are an attractive direction for future work.
>
> **W6:** We will add plots for all datasets in the appendix, observing the same trends for all (also see our response to R1). **W7:** Thank you for catching these! We will fix all citations.
>
> We have additional results that we could not include due to the character limit and are happy to share during the discussion period. We would appreciate the reviewer reconsidering the score in light of the new additions.

---

> > ### Author Rebuttal · Reviewer_DKzj · 2026-04-03
> >
> > I thank the authors for the extensive rebuttal. I read the authors' comment to my review (as well as the experimental results of other reviews). I firmly believe that adding all these additional ablations and the conditional independence analysis would strengthen the paper.
> > Therefore, I am increasing my score.

---

### Official Review · Reviewer_qrJw · 2026-03-13

**Soundness:** 3
**Presentation:** 3
**Significance:** 2
**Originality:** 3
**Overall Recommendation:** 3
**Confidence:** 3

**Summary:**

The paper introduces Self-Augmenting Retrieval for discrete diffusion language models. SARDI leverages the intermediate denoising states  to dynamically update retrieval queries during generation. The model retrieves new evidence to ground uncertain spans before the final response is formed. Experiments on multi-hop QA datasets demonstrate that SARDI outperforms static retrieval and autoregressive baselines in accuracy while offering a 2-6x throughput speedup.

**Compliance With Llm Reviewing Policy:**

Affirmed.

**Final Justification:**

The authors have addressed all of my concerns. However, in their new experiments, a simple ReAct method achieves comparable performance on out domain datasets. Therefore, I increase my score while lowering my confidence.

**Key Questions For Authors:**

* why do you use k = 7?

**Limitations:**

yes

**Strengths And Weaknesses:**

Strengths:

* Combining multi-round retrieval with diffusion denoising is novel.
* The author provides an interesting point on why RAG is particularly compatible with diffusion models. The output tokens are conditionally independent on the retrieved evidence.
* SARDI is more efficient than agentic rag methods.

Weakness:
* The method is worse than Search-R1, and other better and more recent search agent methods are not compared. Although the author claims that these methods "incurs significant engineering overhead", a trivial agentic RAG baseline (a react loop with a retriever) without training is not tested.
* The authors fine-tune DREAM-7B on the combined training sets of 2WikiMultiHopQA and HotpotQA, which is in distribution data. The five benchmarks are all multi-hop and share a lot of similarities, it would be much better if all training data are synthetically generated.

---

> ### Author Rebuttal · Authors · 2026-03-31
>
> We thank the reviewer for the valuable feedback about baselines and in-distribution training.
>
> ### Baselines (W1)
>
> We agree that a training-free agentic baseline is essential and thank the reviewer for this important suggestion. We include Self-RAG (Asai et al, 2024), AdaptiveRAG (Jeong et al, 2024), and a training-free agentic ReAct loop (Yao et al, 2022) with `Retrieve[query]` and `Finish[answer]` actions. All AR methods (except Self-RAG) use Qwen2.5-7B (SFT). We now use each dataset's provided corpus, enabling gold-document recall computation (see response to Reviewer 1) and introducing counterfactual corpora (CofCA, SynthW.) for OOD evaluation.
>
> | | 2Wiki | | HotpotQA | | CofCA | | MuSiQue | | SynthW. | |
> |---|---|---|---|---|---|---|---|---|---|---|
> | Method | EM | T | EM | T | EM | T | EM | T | EM | T |
> | Self-RAG | 15 | 0.42 | 21 | 0.47 | 21 | 0.53 | 5 | 0.54 | 5 | 0.57 |
> | AdaptiveRAG | 34 | 5.37 | 37 | 4.07 | 38 | 2.88 | 15 | 7.38 | 13 | 8.94 |
> | ReAct | 43 | 2.15 | 40 | 2.07 | 43 | 2.02 | 21 | 2.32 | 22 | 3.01 |
> | Search-R1 | 52 | 3.36 | 50 | 2.83 | 44 | 2.94 | 26 | 3.14 | 27 | 3.74 |
> | SARDI K=7 | 58 | 0.39 | 48 | 0.64 | 46 | 0.75 | 20 | 0.88 | 21 | 1.29 |
>
> *T = sec/question*
>
> SARDI outperforms every training-free baseline while being significantly faster. Search-R1 is strongest overall, but SARDI is competitive on 3/5 datasets at 3--8x lower latency without RL training. **We emphasize that SARDI is not intended to replace agentic methods like Search-R1, but to offer an engineering-simple, high-performance RAG alternative with an adjustable quality–latency tradeoff. SARDI is the first method to couple retrieval with intermediate diffusion states.**
>
> ### Finetuning and OOD data (W2)
>
> We agree that it would be ideal to apply SARDI directly to the base diffusion model. Even with few-shot prompts, Dream7B does not consistently output reasoning traces (see below). Since SARDI relies on salient entities in the reasoning trace, it cannot be applied to the Dream7B base model.
>
> | | Correct (w/ reasoning) | Correct (no reasoning) | Wrong |
> |---|---|---|---|
> | Dream7B | 1% | 27% | 72% |
> | Qwen2.5-7B | 32% | 0% | 68% |
> | Dream7B (SFT) | 44% | 0% | 56% |
> | Qwen2.5-7B (SFT) | 44% | 0% | 56% |
>
> Trivedi et al. (2023, ACL) made a directly analogous observation for AR models three years ago:
>
> > *"IRCoT relies on the base LM to have a zero or few-shot CoT-generation ability [...] not as common for small LMs (under 20B) [...] smaller LMs will likely increasingly acquire such ability."*
>
> Their prediction proved correct for AR models; we expect the same for DLMs. Crucially, **after SFT both models reach identical 44% accuracy**, ensuring fair comparison.
>
> - **Controlled SFT.** All SFT baselines use identical training, isolating the retrieval strategy. We note that training on in-domain QA data is common practice (e.g., Search-R1). If the reviewer has a specific synthetic training set in mind, we are happy to run experiments if time permits during the discussion period.
> - **OOD generalization.** CofCA/SynthW. use counterfactual corpora with made-up facts where memorized knowledge cannot help. SARDI is strong on these benchmarks, suggesting gains come from the retrieval mechanism itself.
>
> ### Conditional independence (S2)
>
> We appreciate the reviewer's interest in this insight. To support this empirically, we analyze **conditional mutual information (CMI)**, which measures how coupled two adjacent tokens are: low CMI means the joint distribution is well-approximated by the product of marginals, so tokens can be decoded in parallel. We compute $\text{CMI}(x\_i; x\_{i+1} \mid D) = \mathbb{E}\_{x\_i}[\text{KL}(p(x\_{i+1} \mid x\_i, D) \| p(x\_{i+1} \mid D))]$, approximating the expectation over the top-$k$ logits of $x\_i$, on Dream7B reasoning traces from 2WikiMultiHopQA. We partition by NER into entity and non-entity pairs; roughly half the tokens are named entities.
>
> | | Strong context (all gold) | Gold-1 | Gold-2 | Weak context (no gold) |
> |---|---|---|---|---|
> | CMI (entity) | 0.060 | 0.219 | 0.396 | 0.589 |
> | CMI (non-entity) | 0.121 | 0.199 | 0.257 | 0.286 |
>
> Entity CMI rises 10x as the context quality degrades. This directly supports the conditional independence hypothesis and shows why RAG is particularly suited to parallel decoding.
>
> ### Why K=7?
>
> We chose K=7 as it provides a good accuracy-latency tradeoff. Larger K helps only slightly at the cost of higher latency. See our response to Review 1 for the full ablation across all datasets.
>
> We highlight what we see as the broader contribution of this work: SARDI is the first RAG framework that exploits the denoising trajectory of diffusion language models for progressive query refinement. This is an opportunity unique to diffusion that does not exist in AR models. We hope the new experiments and analysis demonstrate the potential of this direction for simple and efficient retrieval-generation systems, and would appreciate the reviewer's reconsideration of the score.

---

> > ### Author Rebuttal · Reviewer_qrJw · 2026-04-02
> >
> > The authors have addressed all of my concerns. However, in their new experiments, a simple ReAct method achieves comparable performance on out domain datasets. Therefore, I increase my score while lowering my confidence.

---

### Official Review · Reviewer_AmrJ · 2026-03-14

**Soundness:** 3
**Presentation:** 4
**Significance:** 3
**Originality:** 3
**Overall Recommendation:** 4
**Confidence:** 3

**Summary:**

This paper introduces Self-Augmenting Retrieval for Diffusion Language Models (SARDI), a new framework that uses the iterative denoising process of diffusion language models to refine retrieval dynamically during generation. Unlike standard retrieval-augmented generation (RAG) methods that retrieve only once based on the input query, SARDI forms retrieval queries from intermediate diffusion states at different denoising steps and retrieves updated evidence to support later generation. The key idea is that high-confidence tokens can guide retrieval in the early stages, while uncertain parts can be resolved later as more context becomes available. This makes the approach particularly appealing for multi-hop reasoning tasks, where important bridge entities may only emerge during the generation process. Experiments on multi-hop QA benchmarks show that SARDI improves answer accuracy over static retrieval baselines, while also achieving substantially higher throughput than autoregressive models.

**Compliance With Llm Reviewing Policy:**

Affirmed.

**Final Justification:**

I maintain my original recommendation of Weak Accept. The paper explores a novel direction combining diffusion language models with retrieval-augmented generation, with a well-motivated idea and clear writing. The rebuttal satisfactorily addressed my concerns on sampling details, compatibility with Fast-dLLM, and generalization to stronger DLLMs.

**Key Questions For Authors:**

see weaknesses above.

**Limitations:**

As also hinted in the paper’s limitations discussion, an interesting question is whether the method can be extended to stronger existing DLLMs, such as LLaDA 2.0, which might help avoid fine-tuning diffusion language models.

**Strengths And Weaknesses:**

I am not very familiar with the RAG literature, so my comments mainly focus on the novelty, clarity, and the diffusion-language-model side of the work.

strengths

● The idea is novel. To the best of my knowledge, I have not seen much prior work exploring DLLMs in the RAG setting, so this paper studies an interesting and relatively new direction. Parallel generation in DLLMs also seems potentially well suited for retrieval-augmented generation.

● The writing is strong. The paper is easy to follow, and the overall flow from motivation to method to experiments is natural and well organized.

● The empirical results are strong. The experiments suggest that DLLMs can offer an attractive trade-off in the RAG setting: faster generation while still maintaining high answer quality.

weaknesses

● The exploration of DLLM sampling strategies is somewhat limited, and the sampling setup in the experiments feels a bit unclear. I was not always sure about the exact sampling procedure used in practice.

● The paper does not discuss much about whether the decoding process could be further accelerated by combining the method with faster DLLM sampling techniques, such as Fast-dLLM approaches. This seems like a natural direction and would make the paper more complete.

● The paper also does not consider stronger or more recent DLLMs, such as LLaDA 2.0. It would be interesting to discuss whether the proposed framework could be combined with such models, potentially avoiding the need to fine-tuning a diffusion LM.

---

> ### Author Rebuttal · Authors · 2026-03-31
>
> We are grateful for the reviewer's positive assessment and thoughtful suggestions on the diffusion model side.
>
> ### DLLM sampling strategies and parallel decoding (W1)
>
> We appreciate the reviewers interest in the decoding methods and are happy to clarify: We use threshold-based unmasking (Wu et al., 2025a): at each step, all masked positions whose confidence exceeds a threshold ($c_i \geq \tau$) are unmasked and committed to their argmax prediction. Here, the confidence $c\_i = \max\_v p\_\theta(v \mid x\_t, q, D)$ is measured as the softmax probability of the most likely token. If no position exceeds $\tau$, the most confident token is selected to ensure progress. This allows the model to adaptively determine how many tokens to generate per step.
>
> We compare three settings across all datasets:
>
> | | 2Wiki | | HotpotQA | | CofCA | | MuSiQue | | SynthW. | |
> |---|---|---|---|---|---|---|---|---|---|---|
> | Setting | EM | T | EM | T | EM | T | EM | T | EM | T |
> | 50 fixed steps | 59 | 1.33 | 47 | 1.46 | 43 | 1.48 | 20 | 1.42 | 22 | 2.15 |
> | $\tau=0.9$ | 58 | 0.39 | 48 | 0.64 | 46 | 0.75 | 20 | 0.88 | 21 | 1.29 |
> | $\tau=0.95$ | 59 | 0.56 | 49 | 0.90 | 45 | 1.09 | 21 | 1.19 | 21 | 1.78 |
>
> *T = seconds/question.*
>
> Threshold decoding achieves comparable accuracy to fixed-step scheduling while being 2--3x faster. With $\tau=0.9$, SARDI averages 3--7 tokens per step, adapting to question difficulty (6.7 tok/step on 2Wiki, 3.2 on MuSiQue).
>
> We additionally provide new empirical support for why parallel decoding is well-suited to RAG. We analyze **conditional mutual information (CMI)**, which measures how coupled two adjacent tokens are: low CMI means the joint distribution is well-approximated by the product of marginals, so tokens can be decoded in parallel. We compute $\text{CMI}(x\_i; x\_{i+1} \mid D) = \mathbb{E}\_{x\_i}[\text{KL}(p(x\_{i+1} \mid x\_i, D) \| p(x\_{i+1} \mid D))]$, approximating the expectation over the top-$k$ logits of $x\_i$, on Dream7B reasoning traces from 2WikiMultiHopQA. We partition by NER into entity and non-entity pairs; roughly half the tokens overlap with named entities.
>
> | | Strong context (all gold) | Gold-1 | Gold-2 | Weak context (no gold) |
> |---|---|---|---|---|
> | CMI (entity) | 0.060 | 0.219 | 0.396 | 0.589 |
> | CMI (non-entity) | 0.121 | 0.199 | 0.257 | 0.286 |
>
> Entity CMI rises 10x as the context quality degrades. This directly supports the conditional independence hypothesis and shows why RAG is particularly suited to parallel decoding.
>
> ### Combination with Fast-dLLM techniques (W2)
>
> We thank the reviewer for this interesting suggestion. SARDI shares the same threshold-based decoding mechanism as Fast-dLLM. The key additional technique in Fast-dLLM is block diffusion with KV caching, which splits the sequence into blocks and caches the KV states of prefix/suffix blocks across iterations.
>
> The main challenge in dynamic RAG is that retrieved context changes across iterations, requiring re-encoding. However, in SARDI the retrieved set evolves gradually due to smooth query updates from the denoising trajectory:
>
> | | 2Wiki | HotpotQA | CofCA | MuSiQue | SynthW. |
> |---|---|---|---|---|---|
> | SARDI Docs retained/step | 88% | 83% | 89% | 84% | 90% |
>
> Since 83--90% of documents persist between steps, only a small fraction needs recomputation, making document-level KV caching attractive.
>
> We can further combine this with Fast-dLLM by refreshing retrieval every N steps:
>
> | Ret. schedule | 2Wiki EM | HotpotQA EM | CofCA EM | MuSiQue EM |
> |---|---|---|---|---|
> | Every step | 58 | 48 | 46 | 20 |
> | Every 2 steps | 56 | 47 | 45 | 20 |
> | Every 4 steps | 52 | 46 | 45 | 18 |
>
> Between refreshes, the prefix KV cache can be fully reused. This aligns naturally with Fast-dLLM, which already re-encodes cached blocks periodically to correct drift from ignoring bidirectional attention; aligning these re-encoding points with retrieval refreshes adds evidence improvement at no additional cost. This an exciting direction for future work!
>
> ### Stronger DLLMs (W3)
>
> We agree this is an important direction. We chose DREAM-7B because its Qwen2.5-7B initialization enables fair AR comparisons, but SARDI is model-agnostic and can be applied to any RAG-capable DLM. We expect stronger models such as LLaDA 2.0 to further improve performance and potentially reduce or eliminate the need for SFT, as they are more capable of producing structured reasoning via prompting alone. We will clarify this discussion in the paper.
>
> We thank the reviewer again for the helpful suggestions and believe these additions strengthen both the clarity and the diffusion-specific contributions of the paper. We are happy to discuss further during the discussion period.

---

> > ### Author Rebuttal · Reviewer_AmrJ · 2026-04-02
> >
> > Thank you for the detailed rebuttal. The clarifications on the decoding setup, the CMI analysis supporting why parallel decoding is well-suited for RAG, and the discussion on combining SARDI with Fast-dLLM techniques have addressed my main questions well.
> >
> > I will maintain my positive evaluation.

---

### Official Review · Reviewer_dRqw · 2026-03-21

**Soundness:** 3
**Presentation:** 3
**Significance:** 3
**Originality:** 3
**Overall Recommendation:** 4
**Confidence:** 4

**Summary:**

This paper studies how to use diffusion LLMs for RAG. The proposed method leverages confidence-based parallel sampling in diffusion LLMs to enable multi-round iterative retrieval and generation. Experimental results show that on several multi-hop RAG benchmarks, the method outperforms autoregressive LMs and static-retrieval-based RAG approaches.

**Compliance With Llm Reviewing Policy:**

Affirmed.

**Ethical Review Concerns:**

yes.

**Final Justification:**

After reading the authors’ second reply, I believe they have fully addressed my main concerns. In particular, the additional retrieval-level comparisons across methods and retrievers, and the more explicit discussion of the relationship between SARDI, AR Ret@1, and Ret@Adaptive make the empirical picture much clearer.

Taken together, the proposed method appears most compelling as a more efficient and simpler alternative to strong autoregressive iterative retrieval methods, rather than as a method that clearly outperforms the strongest baselines in final answer quality.

I encourage the authors to incorporate these clarifications and additional comparisons into the final version so that the paper’s empirical takeaways and intended positioning are clear to readers.

Accordingly, I have updated my overall recommendation.

**Key Questions For Authors:**

1. Why does AR w/ RET@ADAPTIVE (Jiang et al., 2023) perform worse than AR w/ RET@STATIC and AR w/ RET@N in your experiments?

2. How does the proposed method compare with other representative RAG baselines, such as Self-RAG and Adaptive-RAG, under the same experimental setting?

3. How does SARDI perform when using semantic retrieval models, and how robust is it to variations in retrieval quality and the choice of parameter K?

4. To what extent do the gains come from the proposed iterative retrieval framework itself, rather than from SFT with GPT-4o-mini, especially given the much larger improvements on 2WikiMultihopQA and HotpotQA?

**Limitations:**

yes.

**Strengths And Weaknesses:**

Strengths:
- The paper applies diffusion LLMs to RAG, and the proposed confidence-based iterative RAG framework for multi-hop RAG is interesting and novel.
- The experiments suggest that the proposed method achieves a better performance-efficiency trade-off than autoregressive LM-based approaches.
- The paper is generally well written and easy to follow. The description of the proposed method is clear.

Weaknesses:
- The main issue is that the experimental baselines are not sufficiently comprehensive. In the main experiments, the method is compared only with Jiang et al. (2023) and Search-R1, but not with a broader range of RAG methods such as Self-RAG or Adaptive-RAG. As a result, the experiments do not provide sufficient evidence to fully support the proposed method's effectiveness. This concern is particularly salient because the reported performance of Jiang et al. (2023) is even lower than that of AR w/ RET@10 and AR w/ RET@1 in this paper.
- Although Figure 4 shows that the queries generated by SARDI achieve higher recall than static queries, the paper does not provide sufficiently thorough experiments to demonstrate that the proposed method is better than other baselines in query generation. It also lacks analyses of how the parameter K and the choice of retrieval model affect the overall RAG performance.
- The gains of SARDI on 2WikiMultihopQA and HotpotQA are much more pronounced than those on other out-of-domain datasets, which raises the concern that the main improvements may come from SFT with GPT-4o-mini.

---

> ### Author Rebuttal · Authors · 2026-03-31
>
> We thank the reviewer for the insightful feedback. We have added three RAG baselines, retrieval ablations, and SFT analysis.
>
> ### RET@ADAPTIVE vs. RET@K?
>
> While AR Ret@k uses previous tokens as retrieval queries, RET@ADAPTIVE uses speculatively generated future tokens. In AR models, a single error compounds left-to-right, leading to hallucinated queries that retrieve irrelevant documents. SARDI predicts all token positions in parallel, reducing this effect.
>
> ### Baselines
>
> We appreciate the reviewer's suggestion of a broader set of RAG methods, and include results for Self-RAG (Asai et al., 2024), AdaptiveRAG (Jeong et al., 2024), and a training-free agentic ReAct loop (Yao et al., 2022). All AR methods (except Self-RAG) use Qwen2.5-7B (SFT).
>
> | | 2Wiki | | | | HotpotQA | | | | CofCA | | | | MuSiQue | | | | SynthW. | | | |
> |---|---|---|---|---|---|---|---|---|---|---|---|---|---|---|---|---|---|---|---|---|
> | Method | EM | R | #D | T | EM | R | #D | T | EM | R | #D | T | EM | R | #D | T | EM | R | #D | T |
> | Self-RAG | 15 | 59 | 7 | 0.42 | 21 | 58 | 7 | 0.47 | 21 | 94 | 7 | 0.53 | 5 | 40 | 7 | 0.54 | 5 | 36 | 7 | 0.57 |
> | AdaptiveRAG | 34 | 79 | 13 | 5.37 | 37 | 67 | 12 | 4.07 | 38 | 94 | 11 | 2.88 | 15 | 56 | 13 | 7.38 | 13 | 50 | 13 | 8.94 |
> | ReAct | 43 | 81 | 15 | 2.15 | 40 | 63 | 13 | 2.07 | 43 | 94 | 12 | 2.02 | 21 | 60 | 15 | 2.32 | 22 | 54 | 16 | 3.01 |
> | Search-R1 | 52 | 85 | 16 | 3.36 | 50 | 72 | 14 | 2.83 | 44 | 97 | 13 | 2.94 | 26 | 65 | 16 | 3.14 | 27 | 60 | 18 | 3.74 |
> | AR Ret@1 | 58 | 85 | 20 | 1.27 | 44 | 76 | 21 | 1.39 | 41 | 97 | 16 | 1.37 | 19 | - | 20 | 1.23 | 20 | 55 | 16 | 2.08 |
> | SARDI K=4 | 53 | 75 | 7 | 0.32 | 45 | 69 | 9 | 0.57 | 44 | 95 | 7 | 0.55 | 18 | 56 | 10 | 0.67 | 18 | 45 | 8 | 0.89 |
> | SARDI K=7 | 58 | 83 | 14 | 0.39 | 48 | 74 | 17 | 0.64 | 46 | 97 | 14 | 0.75 | 21 | 63 | 19 | 0.88 | 21 | 53 | 15 | 1.29 |
> | SARDI K=15 | 61 | 90 | 31 | 0.64 | 51 | 81 | 40 | 0.89 | 43 | 98 | 32 | 1.47 | 23 | 72 | 44 | 1.61 | 24 | 62 | 34 | 2.80 |
>
> *R = Cumulative Gold Recall (%), #D = avg. unique docs retrieved, T = seconds/question, K=7 unless specified otherwise*
>
> SARDI outperforms every training-free baseline while being significantly faster. Search-R1 is strongest overall, but SARDI is competitive on 3/5 datasets at 3--8x lower latency without RL training. We emphasize that SARDI is not intended to replace agentic methods like Search-R1, but to offer an engineering-simple, high-performance RAG alternative with an adjustable quality–latency tradeoff. SARDI is the first method to couple retrieval with intermediate diffusion states.
>
> ### Query generation and retrieval
>
> **Retrieval corpora:** For MuSiQue, CofCA, and SynthWorlds, we now use each dataset's provided corpus, enabling gold-document recall computation and introducing counterfactual corpora (CofCA, SynthW.) for out-of-distribution evaluation.
>
> **Query quality.** SARDI achieves high recall, with moderate total documents (see above). Thus, the gains come from query quality, not volume.
>
> **K ablation.** We chose K=7 for a good accuracy-latency tradeoff (see above).
>
> **Dense retriever (E5-base-v2).** SARDI works with both sparse and dense retrievers:
>
> | (EM) | 2Wiki | HotpotQA | CofCA | MuSiQue | SynthW. |
> |---|---|---|---|---|---|
> | Search-R1 (dense) | 41 | 49 | 44 | 26 | 21 |
> | SARDI K=7 (dense) | 50 | 52 | 45 | 22 | 23 |
>
> ### Finetuning and OOD data
>
> We agree that it would be ideal to apply SARDI directly to the base diffusion model.
> Even with few-shot prompts, Dream7B does not reliably output reasoning traces (see below). Since SARDI relies on salient entities surfacing in the reasoning trace, it cannot be applied to the Dream7B base model.
>
> | | Correct (w/ reasoning) | Correct (no reasoning) | Wrong |
> |---|---|---|---|
> | Dream7B | 1% | 27% | 72% |
> | Qwen2.5-7B | 32% | 0% | 68% |
> | Dream7B (SFT) | 44% | 0% | 56% |
> | Qwen2.5-7B (SFT) | 44% | 0% | 56% |
>
> Trivedi et al. (2023, ACL) made a directly analogous observation for AR models three years ago:
>
> > *"IRCoT relies on the base LM to have a zero or few-shot CoT-generation ability [...] not as common for small LMs (under 20B) [...] smaller LMs will likely increasingly acquire such ability."*
>
> Their prediction proved correct for AR models; we expect the same for DLMs. **After SFT both models reach identical 44% accuracy**, ensuring a fair comparison.
>
> 1. **Controlled SFT.** All SFT baselines use identical training, isolating the retrieval strategy. Also, note that the original Search-R1 also trains on in-domain multi-hop QA data via RL.
> 2. **OOD generalization.** CofCA/SynthW. use counterfactual corpora with made-up facts where memorized knowledge cannot help. SARDI is strong on these benchmarks, suggesting the gains come from the retrieval mechanism itself.
>
> We also have additional results that we could not include due to the character limit and are happy to share during the discussion period. We would appreciate the reviewer reconsidering the score in light of the new additions.

---

> > ### Author Rebuttal · Reviewer_dRqw · 2026-04-03
> >
> > Thank you for the rebuttal and the additional experiments. However, I have to admit that I still find it somewhat unclear which specific weaknesses and questions each part of the rebuttal is intended to address.
> >
> > The added baselines are helpful, but they only partially address my concern about the limited comparison. In particular, for Figure 4, I was hoping to see retrieval results from other representative methods under the same setting, rather than only static retrieval, SARDI, and the oracle upper bound. The current rebuttal still does not address this point directly.
> >
> > Similarly, the discussion of retrieval corpora mainly clarifies the updated experimental setup, but does not by itself provide new evidence unless accompanied by more explicit comparisons across different retrieval models.
> >
> > Moreover, I am still not fully convinced by the SFT analysis. It is still unclear to me what the reported 44% accuracy exactly refers to, and I do not think this is the most direct way to isolate the source of the gains. A more convincing comparison would be to train an autoregressive model with the same supervision data and then use it for comparable dynamic query reformulation. That would better clarify how much of the gain comes from the proposed diffusion-based iterative retrieval framework itself.
> >
> > Overall, I appreciate the effort and the additional results, but I still feel that my main concerns are only partially addressed.

---

> > > ### Author Response · Authors · 2026-04-06
> > >
> > > We thank the reviewer for the thoughtful follow-up. We address each of the remaining concerns below:
> > >
> > > ### 1. Document recall across methods and retrievers (Figure 4)
> > > >For Figure 4, I was hoping to see retrieval results from other representative methods under the same setting [...] Similarly, the discussion [...] does not by itself provide new evidence unless accompanied by more explicit comparisons across different retrieval models.
> > >
> > > We thank the reviewer for this suggestion, adding a valuable insight to the paper: We present two metrics that extend Figure 4: **early-stage recall** (at 25% of generation, reflecting query quality early on) and per-query **final recall** (at the last retrieval step), under both BM25 and dense (E5-base-v2) retrievers. See initial rebuttal for document counts, cumulative recall, and EM (also for dense retrievers).
> > >
> > > **Result:** SARDI achieves substantially higher early-stage recall than all baselines **across all retriever types**, indicating that diffusion-based queries surface relevant evidence significantly earlier.
> > >
> > > **2Wiki, Early Recall (25% into generation)**
> > >
> > > ||BM25||||Dense||||
> > > |---|---|---|---|---|---|---|---|---|
> > > |Method|K=3|K=7|K=10|K=15|K=3|K=7|K=10|K=15|
> > > |Search-R1|46|51|53|54|37|39|40|41|
> > > |Ret@Adaptive|55|63|65|67|47|50|51|52|
> > > |AR Ret@1|58|64|67|69|50|53|55|56|
> > > |SARDI (τ=0.9)|63|74|77|80|56|61|63|64|
> > >
> > > **2Wiki, Final Per-Query Recall**
> > >
> > > ||BM25||||Dense||||
> > > |---|---|---|---|---|---|---|---|---|
> > > |Method|K=3|K=7|K=10|K=15|K=3|K=7|K=10|K=15|
> > > |Search-R1|42|53|57|62|28|31|33|35|
> > > |Ret@Adaptive |22|27|29|32|14|17|18|19|
> > > |AR Ret@1 |66|80|85|88|54|61|63|64|
> > > |SARDI (τ=0.9)|67|81|84|88|56|63|64|65|
> > >
> > > **MuSiQue (OOD), Early Recall**
> > >
> > > ||BM25||||Dense||||
> > > |---|---|---|---|---|---|---|---|---|
> > > |Method|K=3|K=7|K=10|K=15|K=3|K=7|K=10|K=15|
> > > |Search-R1|33|38|40|-|33|38|40|42|
> > > |Ret@Adaptive|33|41|45|48|31|37|40|43|
> > > |AR Ret@1|38|46|50|53|41|48|51|55|
> > > |SARDI (τ=0.9)|44|54|59|64|42|52|56|60|
> > >
> > > **MuSiQue (OOD), Final Per-Query Recall**
> > >
> > > ||BM25||||Dense||||
> > > |---|---|---|---|---|---|---|---|---|
> > > |Method|K=3|K=7|K=10|K=15|K=3|K=7|K=10|K=15|
> > > |Search-R1|29|38|42|-|28|35|37|41|
> > > |Ret@Adaptive|14|20|22|25|14|20|21|22|
> > > |AR Ret@1|43|54|58|62|38|48|51|54|
> > > |SARDI (τ=0.9)|44|55|60|64|40|50|53|57|
> > >
> > > **Key observations:**
> > > - SARDI achieves by far the highest early-stage recall across all settings, showing diffusion surfaces key evidence much earlier than AR.
> > > - Final query recall matches or exceeds AR Ret@1 in most settings while running ~3x faster.
> > > - Search-R1's lower per-query recall is expected given its targeted query strategy; its cumulative recall is comparable to SARDI (see initial rebuttal).
> > >
> > > ### 2. SFT analysis
> > >
> > > > It is still unclear to me what the reported 44% accuracy exactly refers to.
> > >
> > > Thank you for pointing this out. The reported 44% refers to Exact Match on 2Wiki under static retrieval (K=7 question-only retrieval), measuring document-understanding independent of dynamic retrieval. The table (initial rebuttal) shows that base Dream7B cannot produce reasoning traces even with few-shot prompting, necessitating SFT.
> > >
> > > **Ruling out SFT as source of the gains:** SFT serves two roles: (1) improving document understanding, and (2) inducing structured reasoning traces. We control for (1) by applying identical SFT to both AR and DLM models. After SFT, both reach identical 44% static RAG accuracy. This ensures that improvements from SARDI are not due to differences in model capability. Role (2) is not fundamental to SFT: SARDI only requires a step-by-step reasoning trace, achievable via few-shot prompting as DLMs mature.
> > >
> > > ### 3. Isolating the diffusion gains: AR with dynamic query formulation
> > >
> > > > A more convincing comparison would be to train an autoregressive model with the same supervision data and then use it for comparable dynamic query reformulation.
> > >
> > > We agree that comparing to an AR model trained with the same supervision and using the same dynamic query formulation is important to isolate the source of gains. Our AR Ret@1 baseline closely provides this comparison: identical SFT and retriever, retrieving at every token using the partial generation as the query (similar to SARDI). We observe that SARDI surfaces evidence earlier, achieves higher EM, and is ~3x faster.
> > >
> > > The only difference to AR Ret@1 is that SARDI can additionally use future token predictions from intermediate diffusion states to inform its queries (which AR models cannot naturally do). To still explore this direction, we include Ret@Adaptive (Jiang et al.), which simulates look-ahead in AR by speculatively generating future sequences to use them as queries. However, AR look-ahead is prone to cascading errors (see initial rebuttal), and the tables confirm this: Diffusion look-ahead is more robust than AR speculation. To our knowledge, this is the closest AR analog to SARDI's iterative retrieval.
> > >
> > > We hope the additions address the remaining concerns. We would appreciate the reviewer reconsidering the score in light of the new results.

---

### Decision · Program_Chairs · 2026-04-30

**Decision:**

Accept (regular)

**Comment:**

- This paper proposes a novel framework (SARDI) that leverages intermediate denoising states in diffusion language models to dynamically refine retrieval during generation. Reviewers generally agreed that coupling retrieval with diffusion decoding is a technically sound and interesting direction, and that the method demonstrates a favorable accuracy–efficiency tradeoff compared to static retrieval and autoregressive baselines.
- During rebuttal, the authors constructively addressed several key concerns by adding stronger RAG baselines (e.g., Self-RAG, AdaptiveRAG, and ReAct), including dense retriever results, and clarifying the role of supervised fine-tuning, which helped improve confidence in the empirical evaluation. However, some limitations remain. In particular, stronger agentic retrieval approaches such as Search-R1 still achieve higher answer quality on some datasets, and the base diffusion model requires fine-tuning to enable reasoning traces, which tempers the “training-free” framing. Additionally, while improved during rebuttal, the evaluation scope remains somewhat limited in model and retriever diversity.
- Overall, the work introduces a meaningful and promising direction for efficient retrieval-augmented generation with non-autoregressive models. Given its novelty and technical soundness despite remaining evaluation limitations, I recommend **Weak Accept**.